# CONSTANT CURVATURE GRAPH CONVOLUTIONAL NETWORKS

## ABSTRACT

Interest has been rising lately towards methods representing data in non-Euclidean spaces, e.g. hyperbolic or spherical, that provide specific inductive biases useful for certain real-world data properties, e.g. scale-free, hierarchical or cyclical. However, the popular *graph neural networks* are currently limited in modeling data only via Euclidean geometry and associated vector space operations. Here, we bridge this gap by proposing mathematically grounded generalizations of *graph convolutional networks* (GCN) to (products of) constant curvature spaces. We do this by i) introducing a unified formalism that can interpolate smoothly between all geometries of constant curvature, ii) leveraging gyro-barycentric coordinates that generalize the classic Euclidean concept of the *center of mass*. Our class of models smoothly recover their Euclidean counterparts when the curvature goes to zero from either side. Empirically, we outperform Euclidean GCNs in the tasks of node classification and distortion minimization for symbolic data exhibiting non-Euclidean behavior, according to their discrete curvature.

## 1 INTRODUCTION

**Graph Convolutional Networks.** The success of convolutional networks and deep learning for image data has inspired generalizations for graphs for which sharing parameters is consistent with the graph geometry. Bruna et al. (2014); Henaff et al. (2015) are the pioneers of spectral graph convolutional neural networks in the graph Fourier space using localized spectral filters on graphs. However, in order to reduce the graph-dependency on the Laplacian eigenmodes, Defferrard et al. (2016) approximate the convolutional filters using Chebyshev polynomials leveraging a result of Hammond et al. (2011). The resulting method (discussed in appendix A) is computationally efficient and superior in terms of accuracy and complexity. Further, Kipf & Welling (2017) simplify this approach by considering first-order polynomials approximations obtaining high scalability. The proposed *graph convolutional networks* (GCN) is interpolating node embeddings via a symmetrically normalized adjacency matrix, while this weight sharing can be understood as an efficient diffusion-like regularizer. Recent works extend GCNs to achieve state of the art results for link prediction (Zhang & Chen, 2018), graph classification (Hamilton et al., 2017; Xu et al., 2018) and node classification (Klicpera et al., 2019; Veličković et al., 2018).

**Euclidean geometry in ML.** In machine learning (ML), data is most often represented in a Euclidean space for various reasons. First, some data is *intrinsically* Euclidean, such as positions in 3D space in classical mechanics. Second, intuition is easier in such spaces, as they possess an appealing vectorial structure allowing basic arithmetic and a rich theory of linear algebra. Finally, a lot of quantities of interest such as distances and inner-products are known in closed-form formulae and can be computed very efficiently on the existing hardware. These operations are the basic building blocks for most of today's popular machine learning models. Thus, the powerful simplicity and efficiency of Euclidean geometry has led to numerous methods achieving state-of-the-art on tasks as diverse as machine translation (Bahdanau et al., 2014; Vaswani et al., 2017), speech recognition (Graves et al., 2013), image classification (He et al., 2016) or recommender systems (He et al., 2017).

**Riemannian ML.** In spite of this success, certain types of data (e.g. hierarchical, scale-free or spherical data) have been shown to be better represented by non-Euclidean geometries (Defferrard et al., 2019; Bronstein et al., 2017; Nickel & Kiela, 2017; Gu et al., 2019), leading in particular to the

rich theories of manifold learning (Roweis & Saul, 2000; Tenenbaum et al., 2000) and information geometry (Amari & Nagaoka, 2007). The mathematical framework in vigor to manipulate non-Euclidean geometries is known as *Riemannian geometry* (Spivak, 1979). Although its theory leads to many strong and elegant results, some of its basic quantities such as the distance function $d(\cdot, \cdot)$ are in general not available in closed-form, which can be prohibitive to many computational methods.

**Representational Advantages of Geometries of Constant Curvature.** An interesting trade-off between general Riemannian manifolds and the Euclidean space is given by manifolds of *constant sectional curvature*. They define together what are called *hyperbolic* (negative curvature), *elliptic* (positive curvature) and Euclidean (zero curvature) geometries. As discussed below and in appendix B, Euclidean spaces have limitations and suffer from large distortion when embedding certain types of data such as trees, e.g. fig. 1. In these cases, the hyperbolic and spherical spaces have representational advantages providing a better inductive bias for the respective data.

The **hyperbolic space** can be intuitively understood as a continuous tree: the volume of a ball grows exponentially with its radius, similarly as how the number of nodes in a binary tree grows exponentially with its depth. Its tree-likeness properties have long been studied mathematically (Gromov, 1987; Hamann, 2017; Ungar, 2008) and it was proven to better embed *complex networks* (Krioukov et al., 2010), *scale-free graphs* and *hierarchical data* compared to the Euclidean geometry (Cho et al., 2019; Sala et al., 2018; Ganea et al., 2018b; Gu et al., 2019; Nickel & Kiela, 2018; 2017; Tifrea et al., 2019). Several important tools or methods found their hyperbolic counterparts, such as variational autoencoders (Mathieu et al., 2019; Ovinnikov, 2019), attention mechanisms (Gulcehre et al., 2018), matrix multiplications, recurrent units and multinomial logistic regression (Ganea et al., 2018a).

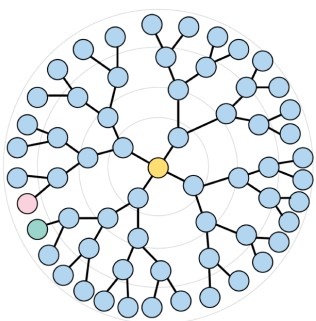

Figure 1: Euclidean space quickly "runs out of space" when fitting exponentially volume growing data such as trees. The embedding distance between the red and the green node keeps decreasing as the number of tree nodes increases, meaning that the graph distance (shortest path length) is no longer accurately represented. Details in appendix B.

Similarly, **spherical geometry** provides benefits for modeling spherical or cyclical data (Defferrard et al., 2019; Matousek, 2013; Davidson et al., 2018; Xu & Durrett, 2018; Gu et al., 2019; Grattarola et al., 2018; Wilson et al., 2014).

**Computational Efficiency of Constant Curvature Spaces (CCS).** CCS are some of the few Riemannian manifolds to possess closed-form formulae for geometric quantities of interest in computational methods, i.e. distance, geodesics, exponential map, parallel transport and their gradients. We also leverage here the closed expressions for weighted centroids.

**"Linear Algebra" of CCS: Gyrovector Spaces.** In order to study the geometry of constant negative curvature in analogy with the Euclidean geometry, Ungar (1999; 2005; 2008; 2016) proposed the elegant non-associative algebraic formalism of **gyrovector spaces**. Recently, Ganea et al. (2018a) have linked this framework to the Riemannian geometry of the space, also generalizing the building blocks for non-Euclidean deep learning models operating with hyperbolic data representations.

However, *it remains unclear how to extend in a principled manner the connection between Riemannian geometry and gyrovector space operations for spaces of constant positive curvature (spherical)*. By leveraging Euler's formula and complex analysis, we present to our knowledge the first unified gyro framework that smoothly interpolates between geometries of constant curvatures irrespective of their signs. This is possible when working with the Poincaré ball and stereographic spherical projection models of respectively hyperbolic and spherical spaces.

### *How should one adapt graph neural networks to non-flat geometries of constant curvature?*

In this work, we propose constant curvature GCNs to model non-Euclidean data. Node embeddings lie in spaces of constant curvature or product of those instead of a Euclidean space, thus leveraging both the representational power of these geometries and the effectiveness of GCNs.

Concurrent to our work, Chami et al. (2019); Liu et al. (2019) propose hyperbolic graph neural networks using tangent space aggregation.

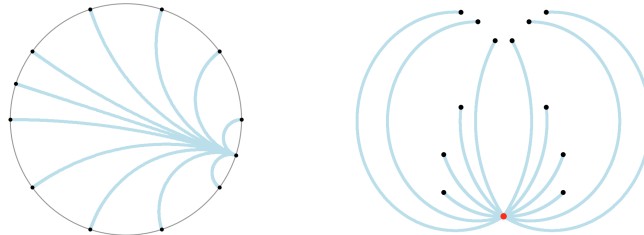

Figure 2: Geodesics in the Poincaré disk (left) and the stereographic projection of the sphere (right).

## 2 THE GEOMETRY OF CONSTANT CURVATURE SPACES

**Riemannian Geometry.** A manifold $\mathcal{M}$ of dimension $d$ is a generalization to higher dimensions of the notion of surface, and is a space that locally *looks* like $\mathbb{R}^d$. At each point $\mathbf{x} \in \mathcal{M}$, $\mathcal{M}$ can be associated a *tangent space* $T_\mathbf{x}\mathcal{M}$, which is a vector space of dimension $d$ that can be understood as a first order approximation of $\mathcal{M}$ around $\mathbf{x}$. A *riemannian metric* $g$ is given by an inner-product $g_\mathbf{x}(\cdot, \cdot)$ at each tangent space $T_\mathbf{x}\mathcal{M}$, $g_\mathbf{x}$ varying smoothly with $\mathbf{x}$. A given $g$ defines the *geometry* of $\mathcal{M}$, because it can be used to define the distance between $\mathbf{x}$ and $\mathbf{y}$ as the infimum of the lengths of smooth paths $\gamma : [0,1] \to \mathcal{M}$ from $\mathbf{x}$ to $\mathbf{y}$, where the length is defined as $\ell(\gamma) := \int_0^1 \sqrt{g_{\gamma(t)}(\dot{\gamma}(t), \dot{\gamma}(t))}\mathrm{d}t$. Under certain assumptions, a given $g$ also defines a *curvature* at each point.

**Unifying all curvatures $\kappa$.** There exist several models of respectively constant positive and negative curvatures. For positive curvature, we choose the stereographic projection of the sphere, while for negative curvature we choose the Poincaré model which is the stereographic projection of the Lorentz model. As explained below, this choice allows us to generalize the gyrovector space framework and unify spaces of both positive and negative curvature $\kappa$ into a single model which we call the $\kappa$-stereographic model.

**The $\kappa$-stereographic model.** For a curvature $\kappa \in \mathbb{R}$ and a dimension $d \geq 2$, it is defined as $\mathfrak{st}_\kappa^d = \{\mathbf{x} \in \mathbb{R}^d \mid -\kappa\|\mathbf{x}\|_2^2 < 1\}$ equipped with its *Riemannian metric* $g_\mathbf{x}^\kappa = \frac{4}{(1+\kappa\|\mathbf{x}\|^2)^2}\mathbf{I} =: (\lambda_\mathbf{x}^\kappa)^2\mathbf{I}$. Note in particular that when $\kappa \geq 0$, $\mathfrak{st}_\kappa^d$ is $\mathbb{R}^d$, while when $\kappa < 0$ it is the open ball of radius $1/\sqrt{-\kappa}$.

**Gyrovector spaces & Riemannian geometry.** As discussed in section 1, the gyrovector space formalism is used to generalize vector spaces to the Poincaré model of hyperbolic geometry (Ungar, 2005; 2008). In addition, important quantities from Riemannian geometry can be rewritten in terms of the Möbius vector addition and scalar-vector multiplication (Ganea et al., 2018a). We here extend gyrovector spaces to the $\kappa$-stereographic model, *i.e.* allowing positive curvature.

For $\kappa > 0$ and any point $\mathbf{x} \in \mathfrak{st}_\kappa^d$, we will denote by $\tilde{\mathbf{x}}$ the unique point of the sphere of radius $\kappa^{-\frac{1}{2}}$ in $\mathbb{R}^{d+1}$ whose stereographic projection is $\mathbf{x}$. As detailed in appendix C.2.2, it is given by

$$\tilde{\mathbf{x}} := (\lambda_\mathbf{x}^\kappa \mathbf{x}, \kappa^{-\frac{1}{2}}(\lambda_\mathbf{x}^\kappa - 1)). \tag{1}$$

For $\mathbf{x}, \mathbf{y} \in \mathfrak{st}_\kappa^d$, we define the $\kappa$-**addition**, in the $\kappa$-stereographic model by:

$$\mathbf{x} \oplus_\kappa \mathbf{y} = \frac{(1 - 2\kappa\mathbf{x}^T\mathbf{y} - \kappa\|\mathbf{y}\|^2)\mathbf{x} + (1 + \kappa\|\mathbf{x}\|^2)\mathbf{y}}{1 - 2\kappa\mathbf{x}^T\mathbf{y} + \kappa^2\|\mathbf{x}\|^2\|\mathbf{y}\|^2} \in \mathfrak{st}_\kappa^d. \tag{2}$$

The $\kappa$-addition is defined in all the cases except for spherical geometry and $\mathbf{x} = \mathbf{y}/(\kappa\|\mathbf{y}\|^2)$ as stated by the following theorem proved in Appendix C.2.1.

**Theorem 1** (Definiteness of $\kappa$-addition). *We have $1 - 2\kappa\mathbf{x}^T\mathbf{y} + \kappa^2\|\mathbf{x}\|^2\|\mathbf{y}\|^2 = 0$ if and only if $\kappa > 0$ and $\mathbf{x} = \mathbf{y}/(\kappa\|\mathbf{y}\|^2)$.*

For $s \in \mathbb{R}$ and $\mathbf{x} \in \mathfrak{st}_\kappa^d$ (and $|s \tan_\kappa^{-1}\|\mathbf{x}\|| < \kappa^{\frac{1}{2}}\pi/2$ if $\kappa > 0$), the $\kappa$-**scaling** in the $\kappa$-stereographic model is given by:

$$s \otimes_\kappa \mathbf{x} = \tan_\kappa\left(s \cdot \tan_\kappa^{-1}\|\mathbf{x}\|\right)\frac{\mathbf{x}}{\|\mathbf{x}\|} \in \mathfrak{st}_\kappa^d, \tag{3}$$

where $\tan_\kappa$ equals $\kappa^{-1/2}\tan$ if $\kappa > 0$ and $(-\kappa)^{-1/2}\tanh$ if $\kappa < 0$. This formalism yields simple closed-forms for various quantities including the distance function inherited from the Riemannian manifold $(\mathfrak{st}_\kappa^d, g^\kappa)$, the exp and log maps, and geodesics, as shown by the following theorem.

**Theorem 2** (Extending gyrovector spaces to positive curvature). *For* $\mathbf{x}, \mathbf{y} \in \mathfrak{st}_\kappa^d$, $\mathbf{x} \neq \mathbf{y}$, $\mathbf{v} \neq \mathbf{0}$, *(and* $\mathbf{x} \neq -\mathbf{y}/(\kappa\|\mathbf{y}\|^2)$ *if* $\kappa > 0$*), the distance function is given by*[a]:

$$d_\kappa(\mathbf{x}, \mathbf{y}) = 2|\kappa|^{-1/2} \tan_\kappa^{-1} \| -\mathbf{x} \oplus_\kappa \mathbf{y}\|, \tag{4}$$

*the unit-speed geodesic from* $\mathbf{x}$ *to* $\mathbf{y}$ *is unique and given by*

$$\gamma_{\mathbf{x} \to \mathbf{y}}(t) = \mathbf{x} \oplus_\kappa (t \otimes_\kappa (-\mathbf{x} \oplus_\kappa \mathbf{y})), \tag{5}$$

*and finally the exponential and logarithmic maps are described as:*

$$\exp_{\mathbf{x}}^\kappa(\mathbf{v}) = \mathbf{x} \oplus_\kappa \left( \tan_\kappa \left( |\kappa|^{\frac{1}{2}} \frac{\lambda_x^\kappa \|\mathbf{v}\|}{2} \right) \frac{\mathbf{v}}{\|\mathbf{v}\|} \right); \log_{\mathbf{x}}^\kappa(\mathbf{y}) = \frac{2|\kappa|^{-\frac{1}{2}}}{\lambda_{\mathbf{x}}^\kappa} \tan_\kappa^{-1} \|{-}\mathbf{x}\oplus_\kappa\mathbf{y}\| \frac{-\mathbf{x} \oplus_\kappa \mathbf{y}}{\| -\mathbf{x} \oplus_k \mathbf{y}\|} \tag{6}$$

---
[a]We write $-\mathbf{x} \oplus \mathbf{y}$ for $(-\mathbf{x}) \oplus \mathbf{y}$ and not $-(\mathbf{x} \oplus \mathbf{y})$.

*Proof sketch:*
The case $\kappa \leq 0$ was already taken care of by (Ganea et al., 2018a). For $\kappa > 0$, we provide a detailed proof in Appendix C.2.2. The exponential map and unit-speed geodesics are obtained using the Egregium theorem and the known formulas in the standard spherical model. The distance then follows from the formula $d_\kappa(\mathbf{x}, \mathbf{y}) = \| \log_{\mathbf{x}}^\kappa(\mathbf{y})\|_{\mathbf{x}}$ which holds in any Riemannian manifold.

$\square$

**Around** $\kappa = 0$. One notably observes that choosing $\kappa = 0$ yields all corresponding Euclidean quantities, which guarantees a *continuous* interpolation between $\kappa$-stereographic models of different curvatures, via Euler's formula $\tan(x) = -i \tanh(ix)$ where $i := \sqrt{-1}$. But is this interpolation *differentiable* with respect to $\kappa$? It is as shown by the following theorem, proved in Appendix C.2.3.

**Theorem 3** (Smoothness of $\mathfrak{st}_\kappa^d$ w.r.t. $\kappa$ around 0). *Let* $\mathbf{v} \neq \mathbf{0}$ *and* $\mathbf{x}, \mathbf{y} \in \mathbb{R}^d$, *such that* $\mathbf{x} \neq \mathbf{y}$ *(and* $\mathbf{x} \neq -\mathbf{y}/(\kappa\|\mathbf{y}\|^2)$ *if* $\kappa > 0$*). Quantities in Eqs. (4,5,6) are well-defined for* $|\kappa| < 1/\min(\|\mathbf{x}\|^2, \|\mathbf{y}\|^2)$, *i.e. for* $\kappa$ *small enough. Their first order derivatives at* $0^-$ *and* $0^+$ *exist and are equal. Moreover, for the distance we have:*

$$d_\kappa(\mathbf{x}, \mathbf{y}) = 2\|\mathbf{x} - \mathbf{y}\| - 2\kappa \left( \|\mathbf{x} - \mathbf{y}\|^3/3 + (\mathbf{x}^T \mathbf{y})\|\mathbf{x} - \mathbf{y}\|^2 \right) + \mathcal{O}(\kappa^2). \tag{7}$$

Note that for $\mathbf{x}^T \mathbf{y} \geq 0$, this tells us that an infinitesimal change of curvature from zero to small negative, *i.e.* towards $0^-$, while keeping $\mathbf{x}, \mathbf{y}$ fixed, has the effect of increasing their distance.

***As a consequence, we have a unified formalism that interpolates smoothly between all three geometries of constant curvature.***

## 3   $\kappa$-GCNs

We start by introducing the methods upon which we build. We present our models for spaces of constant sectional curvature, in the $\kappa$-stereographic model. However, the generalization to cartesian products of such spaces (Gu et al., 2019) follows naturally from these tools.

### 3.1   GRAPH CONVOLUTIONAL NETWORKS

The problem of node classification on a graph has long been tackled with explicit regularization using the graph Laplacian (Weston et al., 2012). Namely, for a directed graph with adjacency matrix $\mathbf{A}$, by adding the following term to the loss: $\sum_{i,j} \mathbf{A}_{ij}\|f(\mathbf{x}_i) - f(\mathbf{x}_j)\|^2 = f(\mathbf{X})^T \mathbf{L} f(\mathbf{X})$, where $\mathbf{L} = \mathbf{D} - \mathbf{A}$ is the (unnormalized) graph Laplacian, $D_{ii} := \sum_k A_{ik}$ defines the (diagonal) degree matrix, $f$ contains the trainable parameters of the model and $\mathbf{X} = (x_i^j)_{ij}$ the node features of the model. Such a regularization is expected to improve generalization if connected nodes in the graph tend to share labels; node $i$ with feature vector $\mathbf{x}_i$ is represented as $f(\mathbf{x}_i)$ in a Euclidean space.

With the aim to obtain more scalable models, Defferrard et al. (2016); Kipf & Welling (2017) propose to make this regularization implicit by incorporating it into what they call *graph convolutional networks* (GCN), which they motivate as a first order approximation of spectral graph convolutions, yielding the following scalable layer architecture (detailed in appendix A):

$$\mathbf{H}^{(t+1)} = \sigma \left( \tilde{\mathbf{D}}^{-\frac{1}{2}} \tilde{\mathbf{A}} \tilde{\mathbf{D}}^{-\frac{1}{2}} \mathbf{H}^{(t)} \mathbf{W}^{(t)} \right), \tag{8}$$

where $\tilde{\mathbf{A}} = \mathbf{A} + \mathbf{I}$ has added self-connections, $\tilde{D}_{ii} = \sum_k \tilde{A}_{ik}$ defines its diagonal degree matrix, $\sigma$ is a non-linearity such as sigmoid, $\tanh$ or $\mathrm{ReLU} = \max(0, \cdot)$, and $\mathbf{W}^{(t)}$ and $\mathbf{H}^{(t)}$ are the parameter and activation matrices of layer $t$ respectively, with $\mathbf{H}^{(0)} = \mathbf{X}$ the input feature matrix.

## 3.2 TOOLS FOR A $\kappa$-GCN

Learning a parametrized function $f_\theta$ that respects hyperbolic geometry has been studied in (Ganea et al., 2018a): neural layers and hyperbolic softmax. We generalize their definitions into the $\kappa$-stereographic model, unifying operations in positive and negative curvature. We explain how curvature introduces a fundamental difference between *left* and *right* matrix multiplications, depicting the *Möbius* matrix multiplication of (Ganea et al., 2018a) as a *right* multiplication, independent for each embedding. We then introduce a *left* multiplication by extension of gyromidpoints which ties the embeddings, which is essential for graph neural networks.

## 3.3 $\kappa$-RIGHT-MATRIX-MULTIPLICATION

Let $\mathbf{X} \in \mathbb{R}^{n \times d}$ denote a matrix whose $n$ rows are $d$-dimensional embeddings in $\mathfrak{st}_\kappa^d$, and let $\mathbf{W} \in \mathbb{R}^{d \times e}$ denote a weight matrix. Let us first understand what a right matrix multiplication is in Euclidean space: the Euclidean right multiplication can be written row-wise as $(\mathbf{XW})_{i\bullet} = \mathbf{X}_{i\bullet} \mathbf{W}$. Hence each $d$-dimensional Euclidean embedding is modified *independently* by a right matrix multiplication. A natural adaptation of this operation to the $\kappa$-stereographic model yields the following definition.

**Definition 1.** *Given a matrix* $\mathbf{X} \in \mathbb{R}^{n \times d}$ *holding $\kappa$-stereographic embeddings in its rows and weights* $\mathbf{W} \in \mathbb{R}^{d \times e}$*, the $\kappa$-**right-matrix-multiplication** is defined row-wise as*

$$(\mathbf{X} \otimes_\kappa \mathbf{W})_{i\bullet} = \exp_0^\kappa ((\log_0^\kappa(\mathbf{X})\mathbf{W})_{i\bullet}) = \tan_\kappa \left( \frac{||(\mathbf{XW})_{i\bullet}||}{||\mathbf{X}_{i\bullet}||} \tan_\kappa^{-1}(||\mathbf{X}_{\bullet} i||) \right) \frac{(\mathbf{XW})_{i\bullet}}{||(\mathbf{XW})_{i\bullet}||}$$

*where* $\exp_0^\kappa$ *and* $\log_0^\kappa$ *denote the exponential and logarithmic map in the $\kappa$-stereographic model.*

This definition is in perfect agreement with the hyperbolic scalar multiplication for $\kappa < 0$, which can also be written as $r \otimes_\kappa \mathbf{x} = \exp_0^\kappa(r \log_0^\kappa(\mathbf{x}))$. This operation is known to have desirable properties such as associativity (Ganea et al., 2018a).

## 3.4 $\kappa$-LEFT-MATRIX-MULTIPLICATION AS A MIDPOINT EXTENSION

For graph neural networks we also need the notion of message passing among neighboring nodes, *i.e.* an operation that *combines / aggregates* the respective embeddings together. In Euclidean space such an operation is given by the left multiplication of the embeddings matrix with the (preprocessed) adjacency $\hat{\mathbf{A}}$: $\mathbf{H}^{(l+1)} = \sigma(\hat{\mathbf{A}} \mathbf{Z}^{(l)})$ where $\mathbf{Z}^{(l)} = \mathbf{H}^{(l)} \mathbf{W}^{(l)}$. Let us consider this left multiplication. For $\mathbf{A} \in \mathbb{R}^{n \times n}$, the matrix product is given row-wise by:

$$(\mathbf{AX})_{i\bullet} = A_{i1} \mathbf{X}_{1\bullet} + \cdots + A_{in} \mathbf{X}_{n\bullet}$$

This means that the new representation of node $i$ is obtained by calculating the linear combination of all the other node embeddings, weighted by the $i$-th row of $\mathbf{A}$. An adaptation to the $\kappa$-stereographic model hence requires a notion of *weighted linear combination*. We propose such an operation in $\mathfrak{st}_\kappa^d$ by performing a $\kappa$-scaling of a *gyromidpoint* $-$ whose definition is reminded below. Indeed, in Euclidean space, the weighted linear combination $\alpha\mathbf{x} + \beta\mathbf{y}$ can be re-written as $(\alpha + \beta)m_{\mathbb{E}}(\mathbf{x}, \mathbf{y}; \alpha, \beta)$ with Euclidean midpoint $m_{\mathbb{E}}(\mathbf{x}, \mathbf{y}; \alpha, \beta) := \frac{\alpha}{\alpha+\beta}\mathbf{x} + \frac{\beta}{\alpha+\beta}\mathbf{y}$. This motivates generalizing the above operation to $\mathfrak{st}_\kappa^d$ as follows.

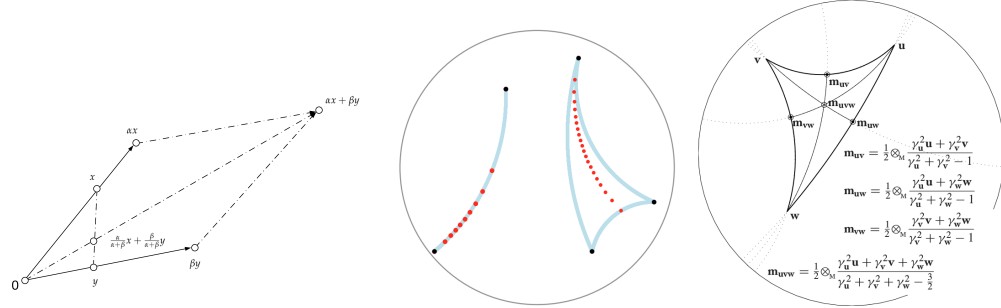

Figure 3: Left: Euclidean Linear combination $\alpha\mathbf{x} + \beta\mathbf{y}$. Middle: Poincaré gyromidpoints (red dots) of two points for different weights on the left and Poincaré gyromidpoints in a hyperbolic triangle on the right with two equal and one free weight. Right: Möbius gyromidpoint in the Poincaré model defined by (Ungar, 2008) and alternatively, here in eq. (10).

**Definition 2.** *Given a matrix* $\mathbf{X} \in \mathbb{R}^{n \times d}$ *holding* $\kappa$-*stereographic embeddings in its rows and weights* $\mathbf{A} \in \mathbb{R}^{n \times n}$, *the* $\kappa$-***left-matrix-multiplication*** *is defined row-wise as*

$$(\mathbf{A} \boxtimes_\kappa \mathbf{X})_{i\bullet} := \left(\sum_j A_{ij}\right) \otimes_\kappa m_\kappa(\mathbf{X}_{1\bullet}, \cdots, \mathbf{X}_{n\bullet}; A_{i1}, \cdots, A_{in}). \tag{9}$$

The $\kappa$-scaling is motivated by the fact that $d_\kappa(\mathbf{0}, r \otimes_\kappa \mathbf{x}) = |r|d_\kappa(\mathbf{0}, \mathbf{x})$ for all $r \in \mathbb{R}$, $\mathbf{x} \in \mathfrak{st}_\kappa^d$. We remind that the gyromidpoint is defined when $\kappa \leq 0$ in the $\kappa$-stereographic model as (Ungar, 2010):

$$m_\kappa(\mathbf{x}_1, \cdots, \mathbf{x}_n; \alpha_1, \cdots, \alpha_n) = \frac{1}{2} \otimes_\kappa \left(\sum_{i=1}^n \frac{\alpha_i \lambda_{\mathbf{x}_i}^\kappa}{\sum_{j=1}^n \alpha_j(\lambda_{\mathbf{x}_j}^\kappa - 1)} \mathbf{x}_i\right), \tag{10}$$

with $\lambda_{\mathbf{x}}^\kappa = 2/(1 + \kappa\|\mathbf{x}\|^2)$. Whenever $\kappa > 0$, we have to further require the following condition:

$$\sum_j \alpha_j(\lambda_{\mathbf{x}_j}^\kappa - 1) \neq 0. \tag{11}$$

For two points, one can calculate that $(\lambda_{\mathbf{x}}^\kappa - 1) + (\lambda_{\mathbf{y}}^\kappa - 1) = 0$ is equivalent to $\kappa\|\mathbf{x}\|\|\mathbf{y}\| = 1$, which holds in particular whenever $\mathbf{x} = -\mathbf{y}/(\kappa\|\mathbf{y}\|^2)$. See fig. 3 for illustrations of gyromidpoints.

Our operation $\boxtimes_\kappa$ satisfies interesting properties, proved in Appendix C.2.4:

**Theorem 4** (Neuter element & $\kappa$-scalar-associativity). *We have* $\mathbf{I}_n \boxtimes_\kappa \mathbf{X} = \mathbf{X}$, *and for* $r \in \mathbb{R}$,

$$r \otimes_\kappa (\mathbf{A} \boxtimes_\kappa \mathbf{X}) = (r\mathbf{A}) \boxtimes_\kappa \mathbf{X}.$$

**The matrix A.** In most graph neural networks, the matrix $\mathbf{A}$ is intented to be a preprocessed adjacency matrix, *i.e.* renormalized by the diagonal degreee matrix $\mathbf{D}_{ii} = \sum_k A_{ik}$. This normalization is often taken either *(i)* to the left: $\mathbf{D}^{-1}\mathbf{A}$, *(ii)* symmetric: $\mathbf{D}^{-\frac{1}{2}}\mathbf{A}\mathbf{D}^{-\frac{1}{2}}$ or *(iii)* to the right: $\mathbf{A}\mathbf{D}^{-1}$. Note that the latter case makes the matrix *right-stochastic*[1], which is a property that is preserved by matrix product and exponentiation. For this case, we prove the following result in Appendix C.2.5:

**Theorem 5** ($\kappa$-left-multiplication by right-stochastic matrices is intrinsic). *If* $\mathbf{A}, \mathbf{B}$ *are right-stochastic,* $\phi$ *is a isometry of* $\mathfrak{st}_\kappa^d$ *and* $\mathbf{X}, \mathbf{Y}$ *are two matrices holding* $\kappa$-*stereographic embeddings:*

$$\forall i, \quad d_\kappa\left((\mathbf{A} \boxtimes_\kappa \phi(\mathbf{X}))_{i\bullet}, (\mathbf{B} \boxtimes_\kappa \phi(\mathbf{Y}))_{i\bullet}\right) = d_\kappa((\mathbf{A} \boxtimes_\kappa \mathbf{X})_{i\bullet}, (\mathbf{B} \boxtimes_\kappa \mathbf{Y})_{i\bullet}). \tag{12}$$

The above result means that $\mathbf{A}$ can easily be preprocessed as to make its $\kappa$-left-multiplication intrinsic to the metric space $(\mathfrak{st}_\kappa^d, d_\kappa)$. At this point, one could wonder: does there exist other ways to take weighted centroids on a Riemannian manifold? We comment on two plausible alternatives.

---

[1]$\mathbf{M}$ is *right-stochastic* if for all $i$, $\sum_j M_{ij} = 1$.

**Fréchet/Karcher means.** They are obtained as $\arg\min_{\mathbf{x}} \sum_i \alpha_i d_\kappa(\mathbf{x}, \mathbf{x}_i)^2$; note that although they are also intrinsic, they usually require solving an optimization problem which can be prohibitively expensive, especially when one requires gradients to flow through the solution − moreover, for the space $\mathfrak{st}_\kappa^d$, it is known that the minimizer is unique if and only if $\kappa \geq 0$.

**Tangential aggregations.** They are defined by lifting the points in a chosen tangent space via the logarithmic map, performing a linear combination and then projecting back via the exponential map, and were in particular used in the recent works of Chami et al. (2019) and Liu et al. (2019). The below theorem describes that for the $\kappa$-stereographic model, this operation is also intrinsic, *i.e.* commutes with isometries. We prove it in Appendix C.2.6.

---

**Theorem 6** (Tangential aggregation is intrinsic). *Define the tangential aggregation of $\mathbf{x}_1, \ldots, \mathbf{x}_n \in \mathfrak{st}_\kappa^d$ w.r.t. weights $\{\alpha_i\}_{1 \leq i \leq n}$, at point $\mathbf{x} \in \mathfrak{st}_\kappa^d$ (for $\mathbf{x}_i \neq -\mathbf{x}/(\kappa\|\mathbf{x}\|^2)$ if $\kappa > 0$) by:*

$$\mathfrak{tg}_{\mathbf{x}}^\kappa(\mathbf{x}_1, ..., \mathbf{x}_n; \alpha_1, ..., \alpha_n) := \exp_{\mathbf{x}}^\kappa \left( \sum_{i=1}^n \alpha_i \log_{\mathbf{x}}^\kappa(\mathbf{x}_i) \right). \tag{13}$$

*For any isometry $\phi$ of $\mathfrak{st}_\kappa^d$, we have*

$$\mathfrak{tg}_{\phi(\mathbf{x})}(\{\phi(\mathbf{x}_i)\}; \{\alpha_i\}) = \phi(\mathfrak{tg}_{\mathbf{x}}(\{\mathbf{x}_i\}; \{\alpha_i\})). \tag{14}$$

---

### 3.5 Logits

Finally, we need the logit and softmax layer, a neccessity for any classification task. We here use the model of (Ganea et al., 2018a), which was obtained in a principled manner for the case of negative curvature. We leave for future work the adaptation of their analysis to positive curvature and use in our experiments the straightforwardly adapted formula to positive curvature, which we detail appendix D.

### 3.6 $\kappa$-GCN

We are now ready to introduce our $\kappa$-stereographic GCN (Kipf & Welling, 2017), denoted by $\kappa$-GCN[2]. Assume we are given a graph with node level features $G = (V, \mathbf{A}, \mathbf{X})$ where $\mathbf{X} \in \mathbb{R}^{n \times d}$ with each row $\mathbf{X}_{i\bullet} \in \mathfrak{st}_\kappa^d$ and adjacency $\mathbf{A} \in \mathbb{R}^{n \times n}$. We first perform a preprocessing step by mapping the Euclidean features to $\mathfrak{st}_\kappa^d$ via the projection $\mathbf{X} \mapsto \mathbf{X}/(2\sqrt{|\kappa|}\|\mathbf{X}\|_{\max})$, where $\|\mathbf{X}\|_{\max}$ denotes the maximal Euclidean norm among all stereographic embeddings in $\mathbf{X}$. For $l \in \{0, \ldots, L-2\}$, the $(l+1)$-th layer of $\kappa$-GCN is given by:

$$\mathbf{H}^{(l+1)} = \sigma^{\otimes_\kappa} \left( \hat{\mathbf{A}} \boxtimes_\kappa \left( \mathbf{H}^{(l)} \otimes_\kappa \mathbf{W}^{(l)} \right) \right), \tag{15}$$

where $\mathbf{H}^{(0)} = \mathbf{X}$, $\sigma^{\otimes_\kappa}(\mathbf{x}) := \exp_{\mathbf{0}}^\kappa(\sigma(\log_{\mathbf{0}}^\kappa(\mathbf{x})))$ is the Möbius version (Ganea et al., 2018a) of a pointwise non-linearity $\sigma$ and $\hat{\mathbf{A}} = \tilde{\mathbf{D}}^{-\frac{1}{2}} \tilde{\mathbf{A}} \tilde{\mathbf{D}}^{-\frac{1}{2}}$. The final layer is a $\kappa$-logit layer (appendix D):

$$\mathbf{H}^{(L)} = \text{softmax} \left( \hat{\mathbf{A}} \, \text{logit}_\kappa \left( \mathbf{H}^{(L-1)}, \mathbf{W}^{(L-1)} \right) \right), \tag{16}$$

where $\mathbf{W}^{(L-1)}$ contains the parameters $\mathbf{a}_k$ and $\mathbf{p}_k$ of the $\kappa$-logits layer. A very important property of $\kappa$-GCN is that its architecture recovers the Euclidean GCN when we let curvature go to zero:

$$\kappa\text{-GCN} \xrightarrow{\kappa \to 0} \text{GCN}.$$

## 4 Experiments

We evaluate the architectures introduced in the previous sections on the tasks of node classification and minimizing embedding distortion for several synthetic as well as real datasets. We detail the training setup and model architecture choices to appendix E.

---

[2]To be pronounced "kappa" GCN; the greek letter $\kappa$ being commonly used to denote sectional curvature

| Model | Tree | Toroidal Graph | Spherical Graph |
|---|---|---|---|
| GCN (Linear) | 0.045 | 0.0607 | 0.0415 |
| GCN (ReLU) | 0.0502 | 0.0603 | 0.0409 |
| $\mathbb{H}^{10}$-GCN | **0.0029** | 0.272 | 0.267 |
| $\mathbb{S}^{10}$-GCN | 0.473 | 0.0485 | **0.0337** |
| $\mathbb{H}^5 \times \mathbb{H}^5$-GCN | 0.0048 | 0.112 | 0.152 |
| $\mathbb{S}^5 \times \mathbb{S}^5$-GCN | 0.51 | **0.0464** | 0.0359 |
| $\left(\mathbb{H}^2\right)^4$- GCN | 0.025 | 0.084 | 0.062 |
| $\left(\mathbb{S}^2\right)^4$- GCN | 0.312 | 0.0481 | 0.0378 |

Table 1: Minimum achieved average distortion of the different models. $\mathbb{H}$ and $\mathbb{S}$ denote hyperbolic and spherical models respectively.

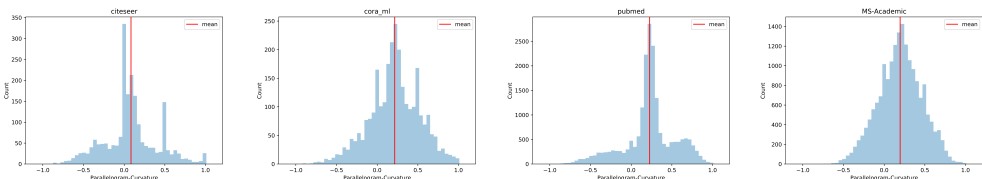

Figure 4: Histogram of Curvatures from "Deviation of Parallogram Law"

**Minimizing Distortion**    Our first goal is to evaluate the graph embeddings learned by our GCN models on the representation task of fitting the graph metric in the embedding space. We desire to minimize the average distortion, i.e. defined similarly as in (Gu et al., 2019): $\frac{1}{n^2} \sum_{i,j} \left( \left( \frac{d(\mathbf{x}_i, \mathbf{x}_j)}{d_G(i,j)} \right)^2 - 1 \right)^2$, where $d(\mathbf{x}_i, \mathbf{x}_j)$ is the distance between the embeddings of nodes i and j, while $d_G(i,j)$ is their graph distance (shortest path length).

We create three synthetic datasets that best reflect the different geometries of interest: i) "Tree'": a balanced tree of depth 5 and branching factor 4 consisting of 1365 nodes and 1364 edges. ii) "Torus": We sample points (nodes) from the (planar) torus, i.e. from the unit connected square; two nodes are connected by an edge iff their toroidal distance (the warped distance) is smaller than a fixed R = 0.01; this gives 1000 nodes and 30626 edges. iii) "Spherical Graph": we sample points (nodes) from $\mathbb{S}^2$, connecting nodes iff their distance is smaller than 0.2, leading to 1000 nodes and 17640 edges.

For the GCN models, we use 1-hot initial node features. We use two GCN layers with dimensions 16 and 10. The non-Euclidean models do not use additional non-linearities between layers. All euclidean parameters are updated using the ADAM optimizer with learning rate 0.01. Curvatures are learned using (stochastic) gradient descent and learning rate of 0.0001. All models are trained for 10000 epochs and we report the minimal achieved distortion. The results shown in table 1 reveal the benefit of our models. One can notice that estimated curvatures correspond to our geometric knowledge about these specific datasets.

### 4.1 NODE CLASSIFICATION

We consider the popular node classification datasets Citeseer (Sen et al., 2008), Cora-ML (McCallum et al., 2000) and Pubmed (Namata et al., 2012). Node labels correspond to the particular subfield the published document is associated with. Dataset statistics and splitting details are deferred to the appendix E due to the lack of space.

**Curvature Estimations of Datasets**    To understand how far are the real graphs of the above datasets from the Euclidean geometry, we first estimate the graph curvature of the four studied datasets using the **deviation from the Parallelogram Law** (Gu et al., 2019) as detailed in appendix F. Curvature histograms are shown in fig. 4. It can be noticed that the datasets are mostly non-Euclidean, thus offering a good motivation to apply our constant-curvature GCN architectures.

| Model | Citeseer | Cora-ML | Pubmed | MS Academic |
|---|---|---|---|---|
| GCN (ReLU) | $75.7 \pm 0.36$ | $83.31 \pm 0.36$ | $\mathbf{79.05 \pm 0.52}$ | $92.14 \pm 0.25$ |
| GCN (Linear) | $\mathbf{76.28 \pm 0.30}$ | $83.81 \pm 0.35$ | $78.94 \pm 0.50$ | $\mathbf{92.3 \pm 0.21}$ |
| $\mathbb{H}^{64}_1$-GCN | $\mathbf{76.29 \pm 0.3}$ | $83.6 \pm 0.34$ | $79.01 \pm 0.58$ | $92.06 \pm 0.22$ |
| $\mathbb{S}^{64}_1$-GCN | $76.18 \pm 0.37$ | $\mathbf{83.97 \pm 0.31}$ | $\mathbf{79.04 \pm 0.5}$ | $92.1 \pm 0.31$ |
| Prod-GCN | $75.91 \pm 0.34$ | $82.9 \pm 0.6$ | $78.7 \pm 0.53$ | $91.9 \pm 0.40$ |

Table 2: Node classification: Average accuracy across 10 splits with estimated uncertainties at 95 percent confidence level via bootstrapping on our datasplits. $\mathbb{H}$ and $\mathbb{S}$ denote hyperbolic (Poincaré ball model) and spherical (stereographic projection) models respectively.

**Training Details**   We trained the Euclidean models with the hyperparameters chosen as reported in (Klicpera et al., 2019). Namely, for GCN we use one hidden layer of size $64$, dropout on the embeddings and the adjacency of rate $0.5$ as well as $L^2$-regularization for the weights of the first layer with $\lambda = 0.02$. Only for Cora-ML we had to adjust the regularization factor $\lambda$ to $0.002$ to ensure similar scores as achieved in (Klicpera et al., 2019).

All Non-Euclidean models use biased-L2 regularization with $\alpha = 10$ and $\lambda = 2e - 2$. Euclidean models used L2 regularization with the same parameter $\lambda$. We used a combination of dropout and dropconnect for the non-Euclidean models. All models have the same number of parameters. We use 2 GCN layers, hidden dimension 64. Product models split hidden dimension into [32, 32] and also input features equally. Non-Euclidean models do not use additional non-linearities. Euclidean parameters use a learning rate of 0.01 for all models using ADAM. The curvatures are learned using gradient descent with a learning rate of 0.01. We show the values of the learned curvatures in appendix E. We use early stopping: we first train for a maximum of 2000 epochs, then we check every 200 epochs for improvement in the validation cross entropy loss; if that is not observed, we stop.

**Node classification results.**   These are shown in table 2. It can be seen that our models are competitive with the two Euclidean GCN considered (with or without non-linearities), showcasing the benefit of our proposed architecture.

## 5   CONCLUSION

In this paper, we introduced a natural extension of graph convolutional networks to the stereographic models of both positive and negative curvatures in a unified manner. We show how this choice of models permits to smoothly interpolate between positive and negative curvature, allowing the curvature of the model to be trained independent of an initial sign choice. We hope that our models will open new exciting directions into non-Euclidean graph neural networks.

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

## A  GCN - A Brief Survey

### A.1  Convolutional Neural Networks on Graphs

One of the pioneering works on neural networks in non-Euclidean domains was done by (Defferrard et al., 2016). Their idea was to extend convolutional neural networks for graphs using tools from **graph signal processing**.

Given a graph $G = (V, \mathbf{A})$, where $\mathbf{A}$ is the adjacency matrix and $V$ is a set of nodes, we define a signal on the nodes of a graph to be a vector $\mathbf{x} \in \mathbb{R}^n$ where $x_i$ is the value of the signal at node $i$. Consider the diagonalization of the symmetrized graph Laplacian $\tilde{\mathbf{L}} = \mathbf{U}\Lambda\mathbf{U}^T$, where $\Lambda = \text{diag}(\lambda_1, \dots, \lambda_n)$. The eigenbasis $\mathbf{U}$ allows to define the graph Fourier transform $\hat{\mathbf{x}} = \mathbf{U}^T\mathbf{x} \in \mathbb{R}^n$.
In order to define a convolution for graphs, we shift from the vertex domain to the **Fourier domain**:

$$\mathbf{x} \star_G \mathbf{y} = \mathbf{U}\left(\left(\mathbf{U}^T\mathbf{x}\right) \odot \left(\mathbf{U}^T\mathbf{y}\right)\right)$$

Note that $\hat{\mathbf{x}} = \mathbf{U}^T\mathbf{x}$ and $\hat{\mathbf{y}} = \mathbf{U}^T\mathbf{y}$ are the graph Fourier representations and we use the element-wise product $\odot$ since convolutions become products in the Fourier domain. The left multiplication with $\mathbf{U}$ maps the Fourier representation back to a vertex representation.
As a consequence, a signal $\mathbf{x}$ filtered by $g_\theta$ becomes $\mathbf{y} = \mathbf{U}g_\theta(\Lambda)\mathbf{U}^Tx$ where $g_\theta = \text{diag}(\theta)$ with $\theta \in \mathbb{R}^n$ constitutes a filter with all parameters free to vary. In order to avoid the resulting complexity $\mathcal{O}(n)$, (Defferrard et al., 2016) replace the non-parametric filter by a polynomial filter:

$$g_\theta(\Lambda) = \sum_{k=0}^{K-1} \theta_k \Lambda^k$$

where $\theta \in \mathbb{R}^K$ resulting in a complexity $\mathcal{O}(K)$. Filtering a signal is unfortunately still expensive since $\mathbf{y} = \mathbf{U}g_\theta(\Lambda)\mathbf{U}^Tx$ requires the multiplication with the Fourier basis $\mathbf{U}$, thus resulting in complexity $\mathcal{O}(n^2)$. As a consequence, (Defferrard et al., 2016) circumvent this problem by choosing the **Chebyshev polynomials** $T_k$ as a polynomial basis, $g_\theta(\Lambda) = \sum_{k=0}^{K} \theta_k T_k(\tilde{\Lambda})$ where $\tilde{\Lambda} = \frac{2\Lambda}{\lambda_{max}} - \mathbf{I}$. As a consequence, the filter operation becomes $\mathbf{y} = \sum_{k=0}^{K} \theta_k T_k(\hat{\mathbf{L}})\mathbf{x}$ where $\hat{\mathbf{L}} = \frac{2\mathbf{L}}{\lambda_{max}} - \mathbf{I}$. This led to a $K$-**localized** filter since it depended on the $K$-th power of the Laplacian. The **recursive** nature of these polynomials allows for an efficient filtering of complexity $\mathcal{O}(K|E|)$, thus leading to an computationally appealing definition of convolution for graphs. The model can also be built in an analogous way to CNNs, by stacking multiple convolutional layers, each layer followed by a non-linearity.

### A.2  Graph Convolutional Networks

(Kipf & Welling, 2017) extended the work of (Defferrard et al., 2016) and inspired many follow-up architectures (Chen et al., 2018; Hamilton et al., 2017; Abu-El-Haija et al., 2018; Wu et al., 2019). The core idea of (Kipf & Welling, 2017) is to limit each filter to 1-hop neighbours by setting $K = 1$, leading to a convolution that is linear in the Laplacian $\hat{\mathbf{L}}$:

$$g_\theta \star \mathbf{x} = \theta_0 \mathbf{x} + \theta_1 \hat{\mathbf{L}}\mathbf{x}$$

They further assume $\lambda_{max} \approx 2$, resulting in the expression

$$g_\theta \star \mathbf{x} = \theta_0 \mathbf{x} - \theta_1 \mathbf{D}^{-\frac{1}{2}}\mathbf{A}\mathbf{D}^{-\frac{1}{2}}\mathbf{x}$$

To additionally alleviate overfitting, (Kipf & Welling, 2017) constrain the parameters as $\theta_0 = -\theta_1 = \theta$, leading to the convolution formula

$$g_\theta \star \mathbf{x} = \theta(\mathbf{I} + \mathbf{D}^{-\frac{1}{2}}\mathbf{A}\mathbf{D}^{-\frac{1}{2}})\mathbf{x}$$

Since $\mathbf{I} + \mathbf{D}^{-\frac{1}{2}}\mathbf{A}\mathbf{D}^{-\frac{1}{2}}$ has its eigenvalues in the range $[0, 2]$, they further employ a reparametrization trick to stop their model from suffering from numerical instabilities:

$$g_\theta \star \mathbf{x} = \theta\tilde{\mathbf{D}}^{-\frac{1}{2}}\tilde{\mathbf{A}}\tilde{\mathbf{D}}^{-\frac{1}{2}}\mathbf{x}$$

where $\tilde{\mathbf{A}} = \mathbf{A} + \mathbf{I}$ and $\tilde{D}_{ii} = \sum_{j=1}^{n} \tilde{A}_{ij}$.

Rewriting the architecture for multiple features $\mathbf{X} \in \mathbb{R}^{n \times d_1}$ and parameters $\Theta \in \mathbb{R}^{d_1 \times d_2}$ instead of $\mathbf{x} \in \mathbb{R}^n$ and $\theta \in \mathbb{R}$, gives

$$\mathbf{Z} = \tilde{\mathbf{D}}^{-\frac{1}{2}} \tilde{\mathbf{A}} \tilde{\mathbf{D}}^{-\frac{1}{2}} \mathbf{X} \Theta \in \mathbb{R}^{n \times d_2}$$

The final model consists of multiple stacks of convolutions, interleaved by a non-linearity $\sigma$:

$$\mathbf{H}^{(k+1)} = \sigma \left( \tilde{\mathbf{D}}^{-\frac{1}{2}} \tilde{\mathbf{A}} \tilde{\mathbf{D}}^{-\frac{1}{2}} \mathbf{H}^{(k)} \Theta^{(k)} \right)$$

where $\mathbf{H}^{(0)} = \mathbf{X}$ and $\Theta \in \mathbb{R}^{n \times d_k}$.

The final output $\mathbf{H}^{(K)} \in \mathbb{R}^{n \times d_K}$ represents the embedding of each node $i$ as $\mathbf{h}_i = \mathbf{H}_{i\bullet} \in \mathbb{R}^{d_K}$ and can be used to perform node classification:

$$\hat{\mathbf{Y}} = \mathrm{softmax} \left( \tilde{\mathbf{D}}^{-\frac{1}{2}} \tilde{\mathbf{A}} \tilde{\mathbf{D}}^{-\frac{1}{2}} \mathbf{H}^{(K)} \mathbf{W} \right) \in \mathbb{R}^{n \times L}$$

where $\mathbf{W} \in \mathbb{R}^{d_K \times L}$, with $L$ denoting the number of classes.

In order to illustrate how embeddings of neighbouring nodes interact, it is easier to view the architecture on the **node level**. Denote by $\mathcal{N}(i)$ the neighbours of node $i$. One can write the embedding of node $i$ at layer $k + 1$ as follows:

$$h_i^{(k+1)} = \sigma \left( \Theta^{(l)} \sum_{j \in \mathcal{N}_i \cup \{i\}} \frac{h_j^{(k)}}{\sqrt{|\mathcal{N}(j)||\mathcal{N}(i)|}} \right)$$

Notice that there is no dependence of the weight matrices $\Theta^{(l)}$ on the node $i$, in fact the same **parameters are shared** across all nodes.

In order to obtain the new embedding $\mathbf{h}_i^{(k+1)}$ of node $i$, we average over all embeddings of the neighbouring nodes. This **Message Passing** mechanism gives rise to a very broad class of graph neural networks (Kipf & Welling, 2017; Veličković et al., 2018; Hamilton et al., 2017; Gilmer et al., 2017; Chen et al., 2018; Klicpera et al., 2019; Abu-El-Haija et al., 2018).

To be more precise, GCN falls into the more general category of models of the form

$$z_i^{(k+1)} = \mathrm{AGGREGATE}^{(k)}(\{h_j^{(k)} : j \in \mathcal{N}(i)\}; W^{(k)})$$
$$h_i^{(k+1)} = \mathrm{COMBINE}^{(k)}(h_i^{(k)}, z_i^{(k+1)}; V^{(k)})$$

Models of the above form are deemed **Message Passing Graph Neural Networks** and many choices for AGGREGATE and COMBINE have been suggested in the literature (Kipf & Welling, 2017; Hamilton et al., 2017; Chen et al., 2018).

## B  GRAPH EMBEDDINGS IN NON-EUCLIDEAN GEOMETRIES

In this section we will motivate non-Euclidean embeddings of graphs and show why the underlying geometry of the embedding space can be very beneficial for its representation. We first introduce a measure of how well a graph is represented by some embedding $f : V \to \mathcal{X}$, $i \mapsto f(i)$:

**Definition 3.** *Given an embedding $f : V \to \mathcal{X}$, $i \mapsto f(i)$ of a graph $G = (V, A)$ in some metric space $\mathcal{X}$, we call $f$ a **D-embedding** for $D \geq 1$ if there exists $r > 0$ such that*

$$r \cdot d_G(i, j) \leq d_{\mathcal{X}}(f(i), f(j)) \leq D \cdot r \cdot d_G(i, j)$$

*The infimum over all such $D$ is called the **distortion** of $f$.*

The $r$ in the definition of distortion allows for scaling of all distances. Note further that a perfect embedding is achieved when $D = 1$.

### B.1  TREES AND HYPERBOLIC SPACE

Trees are graphs that do not allow for a cycle, in other words there is no node $i \in V$ for which there exists a path starting from $i$ and returning back to $i$ without passing through any node twice. The

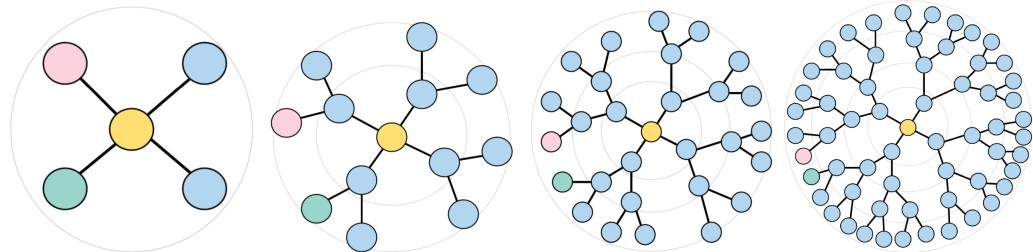

Figure 5: Euclidean embeddings of trees of different depths. All the four most inner circles are identical. Ideal node embeddings should match in distance the graph metric, e.g. the distance between the pink and green nodes should be the same as their shortest path length. Notice how we quickly run out of space, e.g. the pink and green nodes get closer as opposed to farther. This issue is resolved when embedding trees in hyperbolic spaces.

number of nodes increases **exponentially** with the depth of the tree. This is a property that prohibits Euclidean space from representing a tree accurately. What intuitively happens is that "we run out of space". Consider the trees depicted in fig. 5. Here the yellow nodes represent the roots of each tree. Notice how rapidly we struggle to find appropriate places for nodes in the embedding space because their number increases just too fast.

Moreover, graph distances get **extremely distorted** towards the leaves of the tree. Take for instance the green and the pink node. In graph distance they are very far apart as one has to travel up all the way to the root node and back to the border. In Euclidean space however, they are very closely embedded in a $L_2$-sense, hence introducing a big error in the embedding.

This problem can be very nicely illustrated by the following theorem:

**Theorem 7.** *Consider the tree $K_{1,3}$ (also called 3-star) consisting of a root node with three children. Then every embedding $\{x_1, \ldots, x_4\}$ with $x_i \in \mathbb{R}^k$ achieves at least distortion $\frac{2}{\sqrt{3}}$ for any $k \in \mathbb{N}$.*

*Proof.* We will prove this statement by using a special case of the so called **Poincaré-type inequalities** (Deza & Laurent, 1996):

For any $b_1, \ldots, b_k \in \mathbb{R}$ with $\sum_{i=1}^{k} b_i = 0$ and points $x_1, \ldots, x_k \in \mathbb{R}^n$ it holds that

$$\sum_{i,j=1}^{k} b_i b_j ||x_i - x_j||^2 \leq 0$$

Consider now an embedding of the tree $x_1, \ldots, x_4$ where $x_1$ represents the root node. Choosing $b_1 = -3$ and $b_i = 1$ for $i \neq 1$ leads to the inequality

$$||x_2 - x_3||^2 + ||x_2 - x_4||^2 + ||x_3 - x_4||^2 \leq 3||x_1 - x_2||^2 + 3||x_1 - x_3||^2 + 3||x_1 - x_4||^2$$

The left-hand side of this inequality in terms of the graph distance is

$$d_G(2,3)^2 + d_G(2,4)^2 + d_G(3,4)^2 = 2^2 + 2^2 + 2^2 = 12$$

and the right-hand side is

$$3 \cdot d_G(1,2)^2 + 3 \cdot d_G(1,3)^2 + 3 \cdot d_G(1,4)^2 = 3 + 3 + 3 = 9$$

As a result, we always have that the distortion is lower-bounded by $\sqrt{\frac{12}{9}} = \frac{2}{\sqrt{3}}$ □

Euclidean space thus already fails to capture the geometric structure of a very simple tree. This problem can be remedied by replacing the underlying Euclidean space by hyperbolic space.

Consider again the distance function in the Poincaré model, for simplicity with $c = 1$:

$$d_{\mathbb{P}}(x,y) = \cosh^{-1}\left(1 + 2\frac{||x-y||^2}{(1-||x||^2)(1-||y||^2)}\right)$$

Assume that the tree is embedded in the same way as in fig. 5, just restricted to lie in the disk of radius $\frac{1}{\sqrt{c}} = 1$. Notice that as soon as points move closer to the boundary ($||x|| \to 1$), the fraction explodes and the resulting distance goes to infinity. As a result, the further you move points to the border, the more their distance increases, exactly as nodes on different branches are more distant to each other the further down they are in the tree. We can express this advantage in geometry in terms of distortion:

**Theorem 8.** *There exists an embedding $x_1, \ldots, x_4 \in \mathbb{P}^2$ for $K_{1,3}$ achieving distortion $1 + \epsilon$ for $\epsilon > 0$ arbitrary small.*

*Proof.* Since the Poincaré distance is invariant under Möbius translations we can again assume that $x_1 = 0$. Let us place the other nodes on a circle of radius $r$. Their distance to the root is now given as

$$d_{\mathbb{P}}(x_i, 0) = \cosh^{-1}\left(1 + 2\frac{||x_i||^2}{1 - ||x_i||^2}\right) = \cosh^{-1}\left(1 + 2\frac{r^2}{1 - r^2}\right)$$

By invariance of the distance under centered rotations we can assume w.l.o.g. $x_2 = (r, 0)$. We further embed

- $x_3 = \left(r\cos(\frac{2}{3}\pi), r\sin(\frac{2}{3}\pi)\right) = \left(-\frac{r}{2}, \frac{\sqrt{3}}{2}r\right)$

- $x_4 = \left(r\cos(\frac{4}{3}\pi), r\sin(\frac{4}{3}\pi)\right) = \left(-\frac{r}{2}, \frac{\sqrt{3}}{2}r\right)$.

This procedure gives:

$$d_{\mathbb{P}}(x_2, x_3) = \cosh^{-1}\left(1 + 2\frac{||\left(\frac{3r}{2}, \frac{-\sqrt{3}}{2}r\right)||^2}{(1 - r^2)^2}\right) = \cosh^{-1}\left(1 + 2\frac{3r^2}{(1 - r^2)^2}\right)$$

If we let the points now move to the border of the disk we observe that

$$\frac{\cosh^{-1}\left(1 + 2\frac{3r^2}{(1-r^2)^2}\right)}{\cosh^{-1}\left(1 + 2\frac{r^2}{1-r^2}\right)} \xrightarrow{r \to 1} 2$$

But this means in turn that we can achieve distortion $1 + \epsilon$ for $\epsilon > 0$ arbitrary small. QED. $\qquad \square$

The tree-likeliness of hyperbolic space has been investigated on a deeper mathematical level. (Sarkar, 2011) show that a similar statement as in theorem 8 holds for all weighted or unweighted trees. The interested reader is referred to (Hamann, 2017; Sarkar, 2011) for a more in-depth treatment of the subject.

Cycles are the subclasses of graphs that are not allowed in a tree. They consist of one path that reconnects the first and the last node: $(v_1, \ldots, v_n, v_1)$. Again there is a very simple example of a cycle, hinting at the limits Euclidean space incurs when trying to preserve the geometry of these objects (Matousek, 2013).

**Theorem 9.** *Consider the cycle $G = (V, E)$ of length four. Then any embedding $(x_1, \ldots, x_4)$ where $x_i \in \mathbb{R}^k$ achieves at least distortion $\sqrt{2}$.*

*Proof.* Denote by $x_1, x_2, x_3, x_4$ the embeddings in Euclidean space where $x_1, x_3$ and $x_2, x_4$ are the pairs without an edge. Again using the Poincaré-type inequality with $b_1 = b_3 = 1$ and $b_2 = b_4 = -1$ leads to the **short diagonal theorem** (Matousek, 2013):

$$||x_1 - x_3||^2 + ||x_2 - x_4||^2 \leq ||x_1 - x_2||^2 + ||x_2 - x_3||^2 + ||x_3 - x_4||^2 + ||x_4 - x_1||^2$$

The left hand side of this inequality in terms of the graph distance is $d_G(1,3)^2 + d_G(2,4)^2 = 2^2 + 2^2 = 8$ and the right hand side is $1^2 + 1^2 + 1^2 + 1^2 = 4$.
Therefore any embedding has to shorten one diagonal by at least a factor $\sqrt{2}$. $\qquad \square$

It turns out that in spherical space, this problem can be solved perfectly in one dimension for any cycle.

**Theorem 10.** *Given a cycle $G = (V, E)$ of length $n$, there exists an embedding $\{x_1, \ldots, x_n\}$ achieving distortion 1.*

*Proof.* We model the one dimension spherical space as the circle $\mathbb{S}^1$. Placing the points at angles $\frac{2\pi i}{n}$ and using the arclength on the circle as the distance measure leads to an embedding of distortion 1 as all pairwise distances are perfectly preserved. □

Notice that we could also use the exact same embedding in the two dimensional stereographic projection model with $c = 1$ and we would also obtain distortion 1. The difference to the Poincaré disk is that spherical space is finite and the border does not correspond to infinitely distant points. We therefore have no $\epsilon$ since we do not have to pass to a limit.

## C    SPHERICAL SPACE AND ITS GYROSTRUCTURE

Contrarily to hyperbolic geometry, **spherical geometry** is not only in violation with the fifth postulate of Euclid but also with the first. Notice that, shortest paths are not unique as for antipodal (oppositely situated) points, we have infinitely many geodesics connecting the two. Hence the first axiom does not hold. Notice that the third postulate holds as we stated it but it is sometimes also phrased as: "A circle of any center and radius can be constructed". Due to the finiteness of space we cannot have arbitrary large circles and hence phrased that way, the third postulate would not hold.
Finally, we replace the fifth postulate by:

- Given any straight line $l$ and a point $p$ not on $l$, there exists no shortest line $g$ passing through $p$ but never intersecting $l$.

The standard model of spherical geometry suffers from the fact that its underlying space depends directly on the curvature $\kappa$ through a hard constraint $-\kappa \langle \mathbf{x}, \mathbf{x} \rangle = 1$ (similarly to the Lorentz model of hyperbolic geometry). Indeed, when $\kappa \to 0$, the domain diverges to a sphere of infinite radius which is not well defined.
For hyperbolic geometry, we could circumvent the problem by moving to the Poincaré model, which is the stereographic projection of the Lorentz model, relaxing the hard constraint to an inequality. A similar solution is also possible for the spherical model.

### C.1    STEREOGRAPHIC PROJECTION MODEL OF THE SPHERE

In the following we construct a model in perfect duality to the construction of the Poincaré model. Fix the south pole $z = (0, -\frac{1}{\sqrt{\kappa}})$ of the sphere of curvature $\kappa > 0$, *i.e.* of radius $R := \kappa^{-\frac{1}{2}}$. The **stereographic projection** is the map:

$$\Phi : \mathbb{S}_R^n \to \mathbb{R}^n, \mathbf{x}' \mapsto \mathbf{x} = \frac{1}{1 + \sqrt{\kappa}\mathbf{x}'_{n+1}} \mathbf{x}'_{1:n}$$

with the inverse given by

$$\Phi^{-1} : \mathbb{R}^n \to \mathbb{S}_R^n, \mathbf{x} \mapsto \mathbf{x}' = \left( \lambda_{\mathbf{x}}^{\kappa} \mathbf{x}, \frac{1}{\sqrt{\kappa}} (\lambda_{\mathbf{x}}^{\kappa} - 1) \right)$$

where we define $\lambda_{\mathbf{x}}^{\kappa} = \frac{2}{1 + \kappa ||\mathbf{x}||^2}$.
Again we take the image of the sphere $\mathbb{S}_R^n$ under the extended projection $\Phi((0, \ldots, 0, -\frac{1}{\kappa})) = 0$, leading to the stereographic model of the sphere. The metric tensor transforms as:

$$g_{ij}^{\kappa} = (\lambda_{\mathbf{x}}^{\kappa})^2 \delta_{ij}$$

## C.2 GYROVECTOR SPACE IN THE STEREOGRAPHIC MODEL

### C.2.1 PROOF OF THEOREM 1

Using Cauchy-Schwarz's inequality, we have $A := 1 - 2\kappa\mathbf{x}^T\mathbf{y} + \kappa^2||\mathbf{x}||^2||\mathbf{y}||^2 \geq 1 - 2|\kappa|||\mathbf{x}||||\mathbf{y}|| + \kappa^2||\mathbf{x}||^2||\mathbf{y}||^2 = (1 - |\kappa|||\mathbf{x}||||\mathbf{y}||)^2 \geq 0$. Since equality in the Cauchy-Schwarz inequality is only reached for colinear vectors, we have that $A = 0$ is equivalent to $\kappa > 0$ and $\mathbf{x} = \mathbf{y}/(\kappa||\mathbf{y}||^2)$.

### C.2.2 PROOF OF THEOREM 2

Let us start by proving that for $\mathbf{x} \in \mathbb{R}^n$ and $\mathbf{v} \in T_{\mathbf{x}}\mathbb{R}^n$ the **exponential map** is given by

$$\exp_{\mathbf{x}}^{\kappa}(\mathbf{v}) = \frac{\lambda_{\mathbf{x}}^{\kappa}\left(\cos_{\kappa}(\lambda_{\mathbf{x}}^{\kappa}||\mathbf{v}||) - \sqrt{\kappa}\mathbf{x}^T\frac{\mathbf{v}}{||\mathbf{v}||}\sin_{\kappa}(\lambda_{\mathbf{x}}^{\kappa}||\mathbf{v}||)\right)\mathbf{x} + \frac{1}{\sqrt{\kappa}}\sin_{\kappa}(\lambda_{\mathbf{x}}^{\kappa}||\mathbf{v}||)\frac{\mathbf{v}}{||\mathbf{v}||}}{1 + (\lambda_{\mathbf{x}}^{\kappa} - 1)\cos_{\kappa}(\lambda_{\mathbf{x}}^{\kappa}||\mathbf{v}||) - \sqrt{\kappa}\lambda_{\mathbf{x}}^{\kappa}\mathbf{x}^T\frac{\mathbf{v}}{||\mathbf{v}||}\sin_{\kappa}(\lambda_{\mathbf{x}}^{\kappa}||\mathbf{v}||)} \tag{17}$$

Indeed, take a unit speed geodesic $\gamma_{\mathbf{x},\mathbf{v}}(t)$ starting from $\mathbf{x}$ with direction $\mathbf{v}$. Notice that the unit speed geodesic on the sphere starting from $\mathbf{x}' \in \mathbb{S}^{n-1}$ is given by $\Gamma_{\mathbf{x}',\mathbf{v}'}(t) = \mathbf{x}'\cos_{\kappa}(t) + \frac{1}{\sqrt{\kappa}}\sin_{\kappa}(t)\mathbf{v}'$. By the Egregium theorem, we know that $\Phi(\gamma_{\mathbf{x},\mathbf{v}}(t))$ is again a unit speed geodesic in the sphere where $\Phi^{-1} : \mathbf{x} \mapsto \mathbf{x}' = \left(\lambda_{\mathbf{x}}^{\kappa}\mathbf{x}, \frac{1}{\sqrt{\kappa}}(\lambda_{\mathbf{x}}^{\kappa} - 1)\right)$. Hence $\Phi(\gamma_{\mathbf{x},\mathbf{v}}(t))$ is of the form of $\Gamma$ for some $\mathbf{x}'$ and $\mathbf{v}'$. We can determine those by

$$\mathbf{x}' = \Phi^{-1}(\gamma(0)) = \Phi^{-1}(\mathbf{x}) = \left(\lambda_{\mathbf{x}}^{\kappa}\mathbf{x}, \frac{1}{\sqrt{\kappa}}(\lambda_{\mathbf{x}}^{\kappa} - 1)\right)$$

$$\mathbf{v}' = \dot{\Gamma}(0) = \frac{\partial\Phi^{-1}(\mathbf{y})}{\partial\mathbf{y}}\gamma(0)\dot{\gamma}(0)$$

Notice that $\nabla_{\mathbf{x}}\lambda_{\mathbf{x}}^{\kappa} = -\kappa(\lambda_{\mathbf{x}}^{\kappa})^2\mathbf{x}$ and we thus get

$$\mathbf{v}' = \begin{pmatrix} -2\kappa(\lambda_{\mathbf{x}}^{\kappa})^2\mathbf{x}^T\mathbf{v}\mathbf{x} + \lambda_{\mathbf{x}}^{\kappa}\mathbf{v} \\ -\sqrt{\kappa}(\lambda_{\mathbf{x}}^{\kappa})^2\mathbf{x}^T\mathbf{v} \end{pmatrix}$$

We can obtain $\gamma_{\mathbf{x},\mathbf{v}}$ again by inverting back by calculating $\gamma_{\mathbf{x},\mathbf{v}}(t) = \Phi(\Gamma_{\mathbf{x}',\mathbf{v}'}(t))$, resulting in

$$\gamma_{\mathbf{x},\mathbf{v}}(t) = \frac{(\lambda_{\mathbf{x}}^{\kappa}\cos_{\kappa}(t) - \sqrt{\kappa}(\lambda_{\mathbf{x}}^{\kappa})^2\mathbf{x}^T\mathbf{v}\sin_{\kappa}(t))\mathbf{x} + \frac{1}{\sqrt{\kappa}}\lambda_{\mathbf{x}}^{\kappa}\sin_{\kappa}(t)\mathbf{v}}{1 + (\lambda_{\mathbf{x}}^{\kappa} - 1)\cos_{\kappa}(t) - \sqrt{\kappa}(\lambda_{\mathbf{x}}^{\kappa})^2\mathbf{x}^T\mathbf{v}\sin_{\kappa}(t)}$$

Denoting $g_{\mathbf{x}}^{\kappa}(\mathbf{v},\mathbf{v}) = ||\mathbf{v}||^2\lambda_{\mathbf{x}}^{\kappa}$ we have that $\exp_{\mathbf{x}}^{\kappa}(\mathbf{v}) = \gamma_{\mathbf{x},\frac{1}{\sqrt{g_{\mathbf{x}}^{\kappa}(\mathbf{v},\mathbf{v})}}\mathbf{v}}\left(\sqrt{g_{\mathbf{x}}^{\kappa}(\mathbf{v},\mathbf{v})}\right)$ which concludes the proof of the above formula of the exponential map. One then notices that it can be re-written in terms of the $\kappa$-addition. The formula for the logarithmic map is easily checked by verifying that it is indeed the inverse of the exponential map. Finally, the distance formula is obtained via the well-known identity $d_{\kappa}(\mathbf{x},\mathbf{y}) = ||\log_{\mathbf{x}}^{\kappa}(\mathbf{y})||_{\mathbf{x}}$ where $||\mathbf{v}||_{\mathbf{x}} = \sqrt{g_{\mathbf{x}}^{\kappa}(\mathbf{v},\mathbf{v})}$.

Note that as expected, $\exp_{\mathbf{x}}^{\kappa}(\mathbf{v}) \rightarrow_{\kappa\to 0} \mathbf{x} + \mathbf{v}$, converging to the Euclidean exponential map.

$\square$

### C.2.3 PROOF OF THEOREM 3

We first compute a Taylor development of the $\kappa$-addition w.r.t $\kappa$ around zero:

$$\mathbf{x} \oplus_{\kappa} \mathbf{y} = \frac{(1 - 2\kappa\mathbf{x}^T\mathbf{y} - \kappa||\mathbf{y}||^2)\mathbf{x} + (1 + \kappa||\mathbf{x}||^2)\mathbf{y}}{1 - 2\kappa\mathbf{x}^T\mathbf{y} + \kappa^2||\mathbf{x}||^2||\mathbf{y}||^2} \tag{18}$$

$$= [(1 - 2\kappa\mathbf{x}^T\mathbf{y} - \kappa||\mathbf{y}||^2)\mathbf{x} + (1 + \kappa||\mathbf{x}||^2)\mathbf{y}][1 + 2\kappa\mathbf{x}^T\mathbf{y} + \mathcal{O}(\kappa^2)] \tag{19}$$

$$= (1 - 2\kappa\mathbf{x}^T\mathbf{y} - \kappa||\mathbf{y}||^2)\mathbf{x} + (1 + \kappa||\mathbf{x}||^2)\mathbf{y} + 2\kappa\mathbf{x}^T\mathbf{y}[\mathbf{x} + \mathbf{y}] + \mathcal{O}(\kappa^2) \tag{20}$$

$$= (1 - \kappa||\mathbf{y}||^2)\mathbf{x} + (1 + \kappa||\mathbf{x}||^2)\mathbf{y} + 2\kappa(\mathbf{x}^T\mathbf{y})\mathbf{y} + \mathcal{O}(\kappa^2) \tag{21}$$

$$= \mathbf{x} + \mathbf{y} + \kappa[||\mathbf{x}||^2\mathbf{y} - ||\mathbf{y}||^2\mathbf{x} + 2(\mathbf{x}^T\mathbf{y})\mathbf{y}] + \mathcal{O}(\kappa^2). \tag{22}$$

We then notice that using the Taylor of $\|\cdot\|_2$, given by $\|\mathbf{x} + \mathbf{v}\|_2 = \|\mathbf{x}\|_2 + \langle \mathbf{x}, \mathbf{v} \rangle + \mathcal{O}(\|\mathbf{v}\|_2^2)$ for $\mathbf{v} \to \mathbf{0}$, we get

$$\|\mathbf{x} \oplus_\kappa \mathbf{y}\| = \|\mathbf{x} + \mathbf{y}\| + \kappa \langle \|\mathbf{x}\|^2 \mathbf{y} - \|\mathbf{y}\|^2 \mathbf{x} + 2(\mathbf{x}^T \mathbf{y})\mathbf{y}, \mathbf{x} + \mathbf{y} \rangle + \mathcal{O}(\kappa^2) \tag{23}$$

$$= \|\mathbf{x} + \mathbf{y}\| + \kappa(\mathbf{x}^T \mathbf{y})\|\mathbf{x} + \mathbf{y}\|^2 + \mathcal{O}(\kappa^2). \tag{24}$$

Finally Taylor developments of $\tan_\kappa(|\kappa|^{\frac{1}{2}} u)$ and $|\kappa|^{-\frac{1}{2}} \tan_\kappa^{-1}(u)$ w.r.t $\kappa$ around 0 for fixed $u$ yield

$$\text{For } \kappa \to 0^+, \quad \tan_\kappa(|\kappa|^{\frac{1}{2}} u) = \kappa^{-\frac{1}{2}} \tan(\kappa^{\frac{1}{2}} u) \tag{25}$$

$$= \kappa^{-\frac{1}{2}}(\kappa^{\frac{1}{2}} u + \kappa^{\frac{3}{2}} u^3/3 + \mathcal{O}(\kappa^{\frac{5}{2}}) \tag{26}$$

$$= u + \kappa u^3/3 + \mathcal{O}(\kappa^2). \tag{27}$$

$$\text{For } \kappa \to 0^-, \quad \tan_\kappa(|\kappa|^{\frac{1}{2}} u) = (-\kappa)^{-\frac{1}{2}} \tanh((-\kappa)^{\frac{1}{2}} u) \tag{28}$$

$$= (-\kappa)^{-\frac{1}{2}}((-\kappa)^{\frac{1}{2}} u - (-\kappa)^{\frac{3}{2}} u^3/3 + \mathcal{O}(\kappa^{\frac{5}{2}}) \tag{29}$$

$$= u + \kappa u^3/3 + \mathcal{O}(\kappa^2). \tag{30}$$

The left and right derivatives match, hence even though $\kappa \mapsto |\kappa|^{\frac{1}{2}}$ is not differentiable at $\kappa = 0$, the function $\kappa \mapsto \tan_\kappa(|\kappa|^{\frac{1}{2}} u)$ is. A similar analysis yields the same conclusion for $\kappa \mapsto |\kappa|^{-\frac{1}{2}} \tan_\kappa^{-1}(u)$ yielding

$$\text{For } \kappa \to 0, \quad |\kappa|^{-\frac{1}{2}} \tan_\kappa^{-1}(u) = u - \kappa u^3/3 + \mathcal{O}(\kappa^2). \tag{31}$$

Since a composition of differentiable functions is differentiable, we consequently obtain that $\otimes_\kappa$, $\exp^\kappa$, $\log^\kappa$ and $d_\kappa$ are differentiable functions of $\kappa$, under the assumptions on $\mathbf{x}, \mathbf{y}, \mathbf{v}$ stated in Theorem 3. Finally, the Taylor development of $d_\kappa$ follows by composition of Taylor developments:

$$d_\kappa(\mathbf{x}, \mathbf{y}) = 2\|\kappa\|^{-\frac{1}{2}} \tan_\kappa^{-1}(\|(-\mathbf{x}) \oplus_\kappa \mathbf{y}\|)$$

$$= 2(\|\mathbf{x} - \mathbf{y}\| + \kappa((-\mathbf{x})^T \mathbf{y})\|\mathbf{x} - \mathbf{y}\|^2)(1 - (\kappa/3)(\|\mathbf{x} - \mathbf{y}\| + \mathcal{O}(\kappa))^2) + \mathcal{O}(\kappa^2)$$

$$= 2(\|\mathbf{x} - \mathbf{y}\| + \kappa((-\mathbf{x})^T \mathbf{y})\|\mathbf{x} - \mathbf{y}\|^2)(1 - (\kappa/3)\|\mathbf{x} - \mathbf{y}\|^2) + \mathcal{O}(\kappa^2)$$

$$= 2\|\mathbf{x} - \mathbf{y}\| - 2\kappa\left((\mathbf{x}^T \mathbf{y})\|\mathbf{x} - \mathbf{y}\|^2 + \|\mathbf{x} - \mathbf{y}\|^3/3\right) + \mathcal{O}(\kappa^2).$$

$\square$

### C.2.4 PROOF OF THEOREM 4

If $\mathbf{A} = \mathbf{I}_n$ then for all $i$ we have $\sum_j A_{ij} = 1$, hence

$$(\mathbf{I}_n \boxtimes \mathbf{X})_{i\bullet} = \frac{1}{2} \otimes_\kappa \left( \sum_j \frac{\delta_{ij} \lambda_{\mathbf{x}_j}^\kappa}{\sum_k \delta_{ik}(\lambda_{\mathbf{x}_k}^\kappa - 1)} \mathbf{x}_j \right) \tag{32}$$

$$= \frac{1}{2} \otimes_\kappa \left( \frac{\lambda_{\mathbf{x}_i}^\kappa}{(\lambda_{\mathbf{x}_i}^\kappa - 1)} \mathbf{x}_i \right) \tag{33}$$

$$= \frac{1}{2} \otimes_\kappa (2 \otimes_\kappa \mathbf{x}_i) \tag{34}$$

$$= \mathbf{x}_i \tag{35}$$

$$= (\mathbf{X})_{i\bullet}. \tag{36}$$

For associativity, we first note that the gyromidpoint is unchanged by a scalar rescaling of $\mathbf{A}$. The property then follows by scalar associativity of the $\kappa$-scaling.

$\square$

### C.2.5 PROOF OF THEOREM 5

It is proved in (Ungar, 2005) that the gyromidpoint commutes with isometries. The exact same proof holds for positive curvature, with the same algebraic manipulations. Moreover, when the matrix $\mathbf{A}$ is right-stochastic, for each row, the sum over columns gives 1, hence our operation $\boxtimes_\kappa$ reduces to a gyromidpoint. As a consequence, our $\boxtimes_\kappa$ commutes with isometries in this case. Since isometries preserve distance, we have proved the theorem.

$\square$

### C.2.6 PROOF OF THEOREM 6

We begin our proof by stating the *left-cancellation law*:

$$\mathbf{x} \oplus_\kappa (-\mathbf{x} \oplus_\kappa \mathbf{y}) = \mathbf{y} \tag{37}$$

and the following simple identity stating that orthogonal maps commute with $\kappa$-addition

$$\mathbf{R}\mathbf{x} \oplus_\kappa \mathbf{R}\mathbf{y} = \mathbf{R}(\mathbf{x} \oplus_\kappa \mathbf{y}), \quad \forall \mathbf{R} \in O(d) \tag{38}$$

Next, we generalize the gyro operator from Möbius gyrovector spaces as defined in Ungar (2008):

$$\text{gyr}[\mathbf{u}, \mathbf{v}]\mathbf{w} := -(\mathbf{u} \oplus_\kappa \mathbf{v}) \oplus_\kappa (\mathbf{u} \oplus_\kappa (\mathbf{v} \oplus_\kappa \mathbf{w})) \tag{39}$$

Note that this definition applies only for $\mathbf{u}, \mathbf{v}, \mathbf{w} \in \mathfrak{st}_\kappa^d$ for which the $\kappa$-addition is defined (see theorem 1). Following Ungar (2008), we have an alternative formulation (verifiable via computer algebra):

$$\text{gyr}[\mathbf{u}, \mathbf{v}]\mathbf{w} = \mathbf{w} + 2\frac{A\mathbf{u} + B\mathbf{v}}{D}. \tag{40}$$

where the quantities $A, B, D$ have the following closed-form expressions:

$$A = -\kappa^2 \langle \mathbf{u}, \mathbf{w} \rangle \|\mathbf{v}\|^2 - \kappa \langle \mathbf{v}, \mathbf{w} \rangle + 2\kappa^2 \langle \mathbf{u}, \mathbf{v} \rangle \cdot \langle \mathbf{v}, \mathbf{w} \rangle, \tag{41}$$

$$B = -\kappa^2 \langle \mathbf{v}, \mathbf{w} \rangle \|\mathbf{u}\|^2 + \kappa \langle \mathbf{u}, \mathbf{w} \rangle, \tag{42}$$

$$D = 1 - 2\kappa \langle \mathbf{u}, \mathbf{v} \rangle + \kappa^2 \|\mathbf{u}\|^2 \|\mathbf{v}\|^2. \tag{43}$$

We then have the following relations:

**Lemma 11.** *For all $\mathbf{u}, \mathbf{v}, \mathbf{w} \in \mathfrak{st}_\kappa^d$ for which the $\kappa$-addition is defined we have the following relations: i) gyration is a linear map, ii) $\mathbf{u} \oplus_\kappa \mathbf{v} = \text{gyr}[\mathbf{u}, \mathbf{v}](\mathbf{v} \oplus_\kappa \mathbf{u})$, iii) $-(\mathbf{z} \oplus_\kappa \mathbf{u}) \oplus_\kappa (\mathbf{z} \oplus_\kappa \mathbf{v}) = \text{gyr}[\mathbf{z}, \mathbf{u}](-\mathbf{u} \oplus_\kappa \mathbf{v})$, iv) $\|\text{gyr}[\mathbf{u}, \mathbf{v}]\mathbf{w}\| = \|\mathbf{w}\|$.*

*Proof.* The proof is similar with the one for negative curvature given in Ungar (2008). The fact that gyration is a linear map can be easily verified from its definition. For the second part, we have

$$-\text{gyr}[\mathbf{u}, \mathbf{v}](\mathbf{v} \oplus_\kappa \mathbf{u}) = \text{gyr}[\mathbf{u}, \mathbf{v}](-(\mathbf{v} \oplus_\kappa \mathbf{u})) = -(\mathbf{u} \oplus_\kappa \mathbf{v}) \oplus_\kappa (\mathbf{u} \oplus_\kappa (\mathbf{v} \oplus_\kappa (-(\mathbf{v} \oplus_\kappa \mathbf{u})))) = -(\mathbf{u} \oplus_\kappa \mathbf{v}) \tag{44}$$

where the first equality is a trivial consequence of the fact that gyration is a linear map, while the last equality is the consequence of left-cancellation law.

The third part follows easily from the definition of the gyration and the left-cancellation law. The fourth part can be checked using the alternate form in eq. (40). $\square$

We now follow Ungar (2014) and describe all isometries of $\mathfrak{st}_\kappa^d$ spaces:

**Theorem 12.** *Any isometry $\phi$ of $\mathfrak{st}_\kappa^d$ can be uniquely written as:*

$$\phi(\mathbf{x}) = \mathbf{z} \oplus_\kappa \mathbf{R}\mathbf{x}, \quad \text{where } \mathbf{z} \in \mathfrak{st}_\kappa^d, \mathbf{R} \in O(d) \tag{45}$$

The proof is exactly the same as in theorems 3.19 and 3.20 of Ungar (2014), so we will skip it.

We can now prove the main theorem. Let $\phi(\mathbf{x}) = \mathbf{z} \oplus_\kappa \mathbf{R}\mathbf{x}$ be any isometry of $\mathfrak{st}_\kappa^d$, where $\mathbf{R} \in O(d)$ is an orthogonal matrix. Let us denote by $\mathbf{v} := \sum_{i=1}^n \alpha_i \log_\mathbf{x}^\kappa(\mathbf{x}_i)$. Then, using lemma 11 and the formula of the log map from theorem 2, one obtains the following identity:

$$\sum_{i=1}^n \alpha_i \log_{\phi(\mathbf{x})}^\kappa(\phi(\mathbf{x}_i)) = \frac{\lambda_\mathbf{x}^\kappa}{\lambda_{\phi(\mathbf{x})}^\kappa} \text{gyr}[\mathbf{z}, \mathbf{R}\mathbf{x}]\mathbf{R}\mathbf{v} \tag{46}$$

and, thus, using the formula of the exp map from theorem 2 we obtain:

$$\mathfrak{tg}_{\phi(\mathbf{x})}(\{\phi(\mathbf{x}_i)\}; \{\alpha_i\}) = \phi(\mathbf{x}) \oplus_\kappa \mathrm{gyr}[\mathbf{z}, \mathbf{Rx}]\mathbf{R}(-\mathbf{x} \oplus_\kappa \exp_\mathbf{x}^\kappa(\mathbf{v})) \tag{47}$$

Using eq. (39), we get that

$$\mathrm{gyr}[\mathbf{z}, \mathbf{Rx}]\mathbf{Rw} = -\phi(\mathbf{x}) \oplus_\kappa (\mathbf{z} \oplus_\kappa (\mathbf{Rx} \oplus_\kappa \mathbf{Rw})), \quad \forall \mathbf{w} \in \mathfrak{st}_\kappa^d \tag{48}$$

giving the desired

$$\mathfrak{tg}_{\phi(\mathbf{x})}(\{\phi(\mathbf{x}_i)\}; \{\alpha_i\}) = \mathbf{z} \oplus_\kappa \mathbf{R} \exp_\mathbf{x}^\kappa(\mathbf{v}) = \phi(\mathfrak{tg}_\mathbf{x}(\{\mathbf{x}_i\}; \{\alpha_i\})) \tag{49}$$

$$\square$$

## D    Logits

The final element missing in the $\kappa$-GCN is the logit layer, a necessity for any classification task. We here use the formulation of (Ganea et al., 2018a). Denote by $\{1, \ldots, K\}$ the possible labels and let $\mathbf{a}_k \in \mathbb{R}^d$, $b_k \in \mathbb{R}$ and $\mathbf{x} \in \mathbb{R}^d$. The output of a feed forward neural network for classification tasks is usually of the form

$$p(y = k | \mathbf{x}) = \mathrm{softmax}(\langle \mathbf{a}_k, \mathbf{x} \rangle - b_k)$$

In order to generalize this expression to hyperbolic space, the authors of (Ganea et al., 2018a) realized that the term in the softmax can be rewritten as

$$\langle \mathbf{a}_k, \mathbf{x} \rangle - b_k = \mathrm{sign}(\langle \mathbf{a}_k, \mathbf{x} \rangle - b_k)\|\mathbf{a}_k\| d(\mathbf{x}, H_{\mathbf{a}_k, b_k})$$

where $H_{\mathbf{a}, b} = \{\mathbf{x} \in \mathbb{R}^d : \langle \mathbf{x}, \mathbf{a} \rangle - b = 0\} = \{\mathbf{x} \in \mathbb{R}^d : \langle -\mathbf{p} + \mathbf{x}, \mathbf{a} \rangle = 0\} = \tilde{H}_{\mathbf{a}, \mathbf{p}}$ with $\mathbf{p} \in \mathbb{R}^d$. As a first step, they define the hyperbolic hyperplane as

$$\tilde{H}_{\mathbf{a}, \mathbf{p}}^\kappa = \{\mathbf{x} \in \mathfrak{st}_\kappa^d \langle -\mathbf{p} \oplus_\kappa \mathbf{x}, \mathbf{a} \rangle = 0\}$$

where now $\mathbf{a} \in \mathcal{T}_\mathbf{p} \mathfrak{st}_\kappa^d$ and $\mathbf{p} \in \mathfrak{st}_\kappa^d$. They then proceed proving the following formula:

$$d_\kappa(\mathbf{x}, \tilde{H}_{\mathbf{a}, \mathbf{p}}) = \frac{1}{\sqrt{-\kappa}} \sinh^{-1}\left(\frac{2\sqrt{-\kappa}|\langle -\mathbf{p} \oplus_\kappa \mathbf{x}, \mathbf{a} \rangle|}{(1 + \kappa\| - \mathbf{p} \oplus_\kappa \mathbf{x}\|^2)\|\mathbf{a}\|}\right) \tag{50}$$

Using this equation, they were able to obtain the following expression for the logit layer:

$$p(y = k | \mathbf{x}) = \mathrm{softmax}\left(\frac{\|\mathbf{a}_k\|_{\mathbf{p}_k}}{\sqrt{-\kappa}} \sinh^{-1}\left(\frac{2\sqrt{-\kappa}\langle -\mathbf{p}_k \oplus_\kappa \mathbf{x}, \mathbf{a}_k \rangle}{(1 + \kappa\| - \mathbf{p}_k \oplus_\kappa \mathbf{x}\|^2)\|\mathbf{a}_k\|}\right)\right), \tag{51}$$

where $\mathbf{a}_k \in \mathcal{T}_0 \mathfrak{st}_\kappa^d \cong \mathbb{R}^d$, $\mathbf{x} \in \mathfrak{st}_\kappa^d$ and $\mathbf{p}_k \in \mathfrak{st}_\kappa^d$. Combining all these operations leads to the definition of a hyperbolic feed forward neural network. Notice that the weight matrices $\mathbf{W}$ and the normal vectors $\mathbf{a}_k$ live in Euclidean space and hence can be optimized by standard methods such as ADAM (Kingma & Ba, 2015).

For positive curvature $\kappa > 0$ we use in our experiments the following formula for the softmax layer:

$$p(y = k | \mathbf{x}) = \mathrm{softmax}\left(\frac{\|\mathbf{a}_k\|_{\mathbf{p}_k}}{\sqrt{\kappa}} \sin^{-1}\left(\frac{2\sqrt{\kappa}\langle -\mathbf{p}_k \oplus_\kappa \mathbf{x}, \mathbf{a}_k \rangle}{(1 + \kappa\| - \mathbf{p}_k \oplus_\kappa \mathbf{x}\|^2)\|\mathbf{a}_k\|}\right)\right), \tag{52}$$

which is inspired from the formula $i \sin(x) = \sinh(ix)$ where $i := \sqrt{-1}$. However, we leave for future work the rigorous proof that the distance to geodesic hyperplanes in the positive curvature setting is given by this formula.

## E    More Experimental Details

We here present training details for the node classification experiments.

We closely follow the training and evaluation scheme from previous work, e.g. (Klicpera et al., 2019). We split the data into training, early stopping, validation and test set. Namely we first split the dataset into a known subset of size $n_{known}$ and an unknown subset consisting of the rest of the nodes. For

| Dataset | Type | Classes | Features | Nodes | Edges | Label rate | Avg sp |
|---------|------|---------|----------|-------|-------|-----------|--------|
| Citeseer | Citation | 6 | 3703 | 2110 | 3668 | 0.036 | 9.31 |
| Cora-ML | Citation | 7 | 2879 | 2810 | 7981 | 0.047 | 5.27 |
| Pubmed | Citation | 3 | 500 | 19717 | 44324 | 0.003 | 6.34 |
| MS-Academic | Co-author | 15 | 6805 | 18333 | 81894 | 0.0016 | 5.34 |

Table 3: Summary statistics for the four datasets, where sp denotes shortest path.

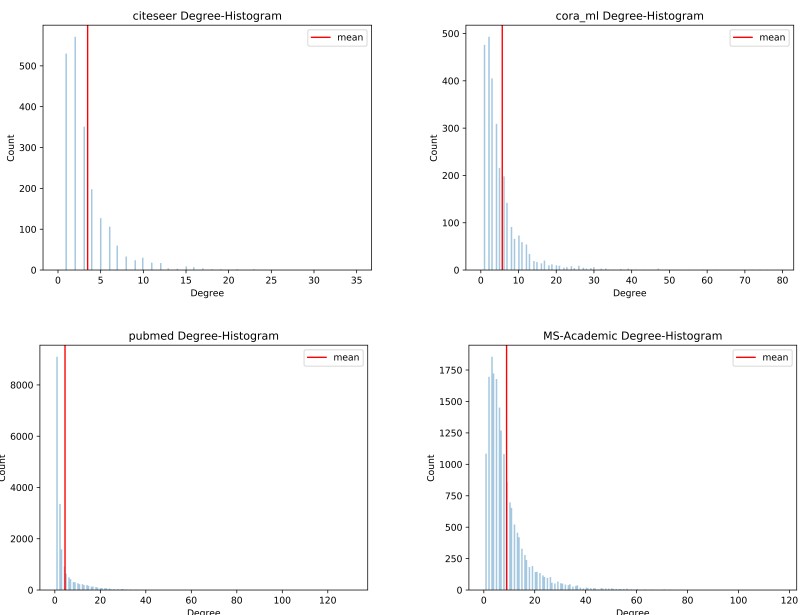

Figure 6: Histogram of node degrees

all the graphs we use $n_{known} = 1500$ except for MS Academics, where we use $n_{known} = 5000$. The known subset is further split into a training set consisting of 20 data points per label, an early stopping set of size 500 and a validation set of the remaining nodes. Notice that the whole structure of the graph and all the node features are used in an unsupervised fashion since the embedding of a training node might for instance depend on the embedding of a node from the validation set. But when calculating the loss, we only provide supervision with the training data.

The unknown subset serves as the test data and is only used for the final evaluation of the model. Hyperparameter-tuning is performed on the validation set. We further use early stopping in all the experiments. We stop training as soon as the early stopping cross entropy loss has not decreased in the last $n_{patience} = 200$ epochs or as soon as we have reached $n_{max} = 2000$ epochs. The model chosen is the one with the highest accuracy score on the early stopping set. For the final evaluation we test the model on 10 different data splits and report mean accuracy and bootstrapped confidence intervals. We use the described setup for both the Euclidean and non-Euclidean models to ensure a fair comparison.

**Learned curvatures** .
Citeseer:
Hyp GCN: Trained curvature, average curvature over all runs: -1.057 +-0.03
Sphr GCN: Trained curvature, average curvature over all runs: 0.951 +-0.019
Prod GCN: Trained curvatures, average curvatures: [1.331, -0.91]

Cora
Hyp GCN: Trained curvature, average curvature: -1.127 +-0.011
Sphr GCN: Trained curvature, average curvature: 0.857+-0.013

Prod GCN: Trained curvature, average curvature: [-1.03, -1.01]

Pubmed:
Hyp GCN: Trained curvature, average curvature: 1.123 +- 0.01
Sphr GCN: Trained curvature, average curvature: 0.896 +- 0.008
Prod GCN: Not training curvature, fixed to [-1, -1]

Ms-Academic
Hyp GCN: Trained curvature, average curvature: 1.26 +- 0.09
Sphr GCN: Trained curvature, average curvature: 0.8 +- 0.07
Prod GCN: Not training curvature, fixed to [-1, -1]

## F GRAPH CURVATURE ESTIMATION ALGORITHM

We used the following procedure to estimate the curvature of a dataset developed by Gu et al. (2019):

1. Fix a node $m \in G$ and sample two neighbouring nodes $a, b \in G$ uniformly. Further sample an additional reference node $c \in G$ uniformly (again avoiding $m = c$).

2. Calculate $\psi(m; a, b; c) = \frac{1}{2d_G(a,b)} \left( d_G^2(a, m) + \frac{d_G^2(b,c)}{4} - \left( \frac{d_G^2(a,b) + d_G^2(a,c)}{2} \right) \right)$

3. Reiterate the above sampling $n_{iter}$ times and obtain an average curvature at node $m$.

4. Do this procedure for every node $m \in G$.

