# OpenReview forum: "Constant Curvature Graph Convolutional Networks"
_ICLR.cc/2020/Conference — Reject_

### Official Review · AnonReviewer1 · 2019-10-16
**Official Blind Review #1**

**Rating:** 1

**Review:**

This paper builds a new graph convolutional network (GCN) based on hyperbolic representations of the graph nodes: all latent representations of the nodes are points on Poincare disks. The authors adapted the Hyperbolic Neural Networks by Ganea et al. (2018) into the Kipf & Welling's (2016) version of GCN. Specifically, the authors variated the right matrix multiplication in GCN with Ganea et al.'s Mobius matrix-vector multiplication (that can be regarded as a transformation between two Poincare disks). Moreover, as a non-trivial adaptation, the author defined the left matrix multiplication in GCN (that can be regarded as a weighted linear combination of several points on the same Poincare disk) with Ungar's (2010) weighted barycenter, which is from computational geometry but not the machine learning community. The resulting method is tested on a toy problem and semi-supervised node classification of commonly used datasets, showing the possibility of improvement.

This is a practical contribution but not a theoretical contribution, as there is no main theorem (or equivalent statements), and the main novelty is on using Ungar's (2010) weighted barycenter to perform neighborhood features aggregations. There are some general discussions on how to adapt gyrovector space theory into spherical spaces. This is interesting but no formal results are presented.

I vote for rejection for four major weaknesses explained as follows.

(1) The experimental results cannot show the usefulness of the proposed GCN. The results on real datasets are similar to the regular GCN. As the authors themselves remarked, "it can be seen that our models sometimes outperform the two Euclidean GCN". The experimental settings, e.g. how the train:test datasets are split, and hyperparameter settings, are not clearly given.

(2) The method is not well motivated. The motivation, based on the writing, is that constant curvature spaces are more general where the computation is easy to handle. This is too general and not enough as a motivation. After reading, the reviewer cannot understand why the user should bother to use hyperbolic representations that are more complex to compute in GCNs, given that the experimental results are roughly the same.

(3) A large body of graph neural network literature is omitted. The authors start from a very high-level description of machine learning in constant curvature spaces. Such high-level introductions require more comprehensive literatures to support. In the first mentioning of Graph Neural Networks, the authors only cited Kipf & Welling's GCN. This is misleading. For example, the original GNN, ChebNet, etc. that leads to GCN can be mentioned. The reviewer recommends the authors to read some literature review of the topic of GCN and re-organize the references, and use search engines to have a better view on the state of the art of (hyperbolic) geometric theories and graph convolutional networks. This is a thriving area that requires a careful literature review.

(4) The writing quality is not satisfactory. Here are a few examples: The ICLR citation style needs to use sometimes \citep. The authors instead used \cite everywhere, making the paper hard to read. The authors are suggested to use unified notations to denote vectors/matrices (e.g. all bold). The introduction can start at a lower level (such as flat/hyperbolic neural networks). Section 3.4, as the main technical innovation, can be extended and includes some demonstrations.

**Experience Assessment:**

I have published one or two papers in this area.

**Review Assessment: Checking Correctness Of Derivations And Theory:**

I assessed the sensibility of the derivations and theory.

**Review Assessment: Checking Correctness Of Experiments:**

I carefully checked the experiments.

**Review Assessment: Thoroughness In Paper Reading:**

I read the paper at least twice and used my best judgement in assessing the paper.

---

> ### Author Response · Authors · 2019-11-11
> **Paper improvement**
>
> We would like to thank reviewer #1 for the constructive critics.
>
> We have significantly improved the exposition in this paper, ameliorated the discussion of the related work, added new theoretical results in the main text, compressed the appendix and enhanced the description of the experimental setup. We have updated the paper pdf and kindly ask the reviewer to take another look.
>
> THEORY:
> Section 2 now formalizes our contribution of unifying Riemannian geometry and gyrovector spaces for all stereographic spaces of constant curvature of any sign, formally proving that the interpolation is smooth at 0. Please note that gyrovector spaces and their properties were stated only for negative curvature spaces prior to our work.
> Theorem 1 formally states when k-addition is defined for spherical spaces.
> Theorem 2 formally derives formulas of exponential map, logarithmic map, geodesics and distance w.r.t. Gyro-operations in the setting of positive curvature, by making use of the Egregium theorem.
> Theorem 3 proves smoothness of the main quantities appearing in Theorem 2 w.r.t. curvature, i.e. that they are differentiable at 0, by showing that left and right derivatives from both models actually match. It also gives the derivative of the distance function w.r.t. curvature around 0.
> Theorem 4 describes two simple properties of our proposed left-matrix-multiplication, in particular that it is Mobius-scalar-associative.
> Theorem 5 shows that our proposed left-matrix-multiplication is an intrinsic averaging for Riemannian manifolds of constant curvature, i.e. it does commute with isometries when the matrix A is right-stochastic, which is not the case of right-matrix-multiplication. This is a desirable property for Riemannian averaging.
> Theorem 6 shows that weighted combinations in the tangent space used in the very recent works of [1,2] (appeared after ICLR submission deadline) is also an intrinsic averaging for Riemannian manifolds of constant sectional curvature, i.e. it does commute with isometries.
>
> LITERATURE:
> A more careful review of the literature has been incorporated into the introduction section.
> We have also incorporated the recent works [1,2] to appear soon at Neurips’19 (their text became available very recently and after the ICLR submission deadline).
>
> WRITING:
> We have unified all notations: bold for multidimensional quantities, i.e. vectors, matrices, embeddings.
> We have significantly improved the writing, re-done the bibliography and citing, and organized the most important theorems and definitions into a clearer presentation.
>
> MOTIVATION:
> We have rewritten the introduction and detailed the motivation of using spaces of constant curvature, i.e. they better represent certain classes of data such as hierarchical structures, scale-free graphs, complex networks and cyclical or spherical data. This is in accordance with the previous related work on non-Euclidean embeddings. Moreover, we have detailed some limitations of Euclidean spaces in Appendix B.
>
> EXPERIMENTAL SETTINGS & HYPERPARAMETERS:
> Are added to section 4 and appendix E
>
> EXPERIMENTAL RESULTS:
> Our methods outperform the Euclidean GCN models for the synthetic datasets for minimizing distortion, while being better or competitive for node classification. We believe further experimental investigations are needed to better train non-Euclidean graph neural network models.
>
>
> [1] Hyperbolic Graph Neural Networks, I. Chami et al., Neurips’19
> [2] Hyperbolic Graph Convolutional Networks, Q. Liu et al., Neurips’19

---

> > ### Author Response · Authors · 2019-11-14
> > **Thank you for taking another look at our improved paper**
> >
> > We would love to hear your feedback on our substantially improved paper (based on your suggestions). Unfortunately, ICLR's tight schedule would prevent us to answer any additional questions after 15th of November. Thank you!

---

### Official Review · AnonReviewer3 · 2019-10-23
**Official Blind Review #3**

**Rating:** 8

**Review:**

Summary:
The authors propose using non-Euclidean spaces for GCNs. This is inspired by the recent work into non-Euclidean, and especially hyperbolic, embeddings. A few papers have recently tried to go past embeddings into building non-Euclidean models, requiring the lifting of standard operations in Euclidean space to non-Euclidean settings. This has been done in particular in hyperbolic space, but some datasets benefit from more complex spaces. The authors combine the mixed-curvature product formalism that uses products of Euclidean, hyperbolic, and spherical spaces for embeddings, but use these for GCN operations.

Doing this requires, in particular, developing a reasonable way to perform these operations in spherical space (since Euclidean is trivial and hyperbolic has been recently worked on). The authors do a nice lifting via complex operations, and both the hyperbolic and spherical spaces can devolve into the flat Euclidean space when their curvature goes to 0. The authors implement these GCNs, train the curvatures, and demonstrate performance improvements over Euclidean only versions on node classification on benchmark datasets. They also give a fairly nice introduction to all of these ideas in an extended appendix.


Strengths, Weaknesses, Recommendation:
This paper is reasonably interesting---it joins an effort to produce non-Euclidean models in a tractable way, which is fairly challenging, but could have a good impact. On the plus side, it's great that the authors added the nice development for the spherical operations, since that will come in handy for many models. The experiments are also good. On the downside, everything here is an extension of existing work, and the body of the paper is hard to read (though this may be inevitable, there's a lot of background to go over here). Overall I recommend accepting it; I think it's a solid contribution.



Comments:
- I don't understand why the authors say that their space "interpolates smoothly" just because the limit in the curvature is the same from the left and right side. For example, the absolute value function has the same limit from the left and the right at 0, but it's not differentiable there. Is it actually true that if we take the derivatives of the piecewise hyperbolic/spherical distance function that it's differentiable at c=0?

- There are a couple of recent papers that also consider hyperbolic GCNs, and in fact use  similar ideas for the aggregation and update steps (i.e., same lift to hyperbolic space). However, these were recently NeurIPS papers, and the text is not yet out, so I don't think this should affect the authors' independent work (and also the product part is new). I do recommend that the authors compare against those results in a future update of this work. The papers are "Hyperbolic Graph Convolutional Neural Networks" by Chami et al and Hyperbolic Graph Neural Networks by Liu et al.

- One thing that I didn't see discussed by the authors is that there are subtle difference between hyperbolic and spherical spaces. For example, the weighted midpoint of Def. 3.2 doesn't immediately extend to spherical space (or at least won't be unique). As an example, consider S^2 and the mean of two antipodal points on it---there's many choices for the midpoint. You probably have to limit the operation to a half-sphere (there's some ideas for this in Gu et al).

- For the synthetic tree, why is the number of edges 2(|V|-1) rather than |V|-1?

- Are the curvatures the same for each layer for the GCNs? This is an interesting point to discuss (some of the NeurIPS papers I mentioned train the curvature for each layer). Also, how do you select the number of factors of each type?

- Minor, but some of these citations can be updated. The "De Sa" et al 2018 arxiv citation is really Sala et al and is an ICML '18 paper. Similarly, Gulcehre et al is a 2019 ICLR paper, and so on. It's always good to get these right.

- Is there any actual empirical importance from recovering the Euclidean case exactly for 0 curvature? The reason I ask is that my experience is that the hyperboloid is typically easier to work with.

- One useful thing to point out in B.3.3 is that in general, it need not be a diffeomorphism for all of M for any manifold, which leads to non-uniqueness. In differential geometry, the "cut locus" is the region beyond which there is this non-uniqueness.

- In the appendix, the statement "Sarkar (2011) show that a similar statement as in Theorem 2 holds for a very general class of trees" is confusing to me. The "general class", as far as I know, is actually *all* trees, weighted or unweighted.

**Experience Assessment:**

I have published one or two papers in this area.

**Review Assessment: Checking Correctness Of Derivations And Theory:**

I assessed the sensibility of the derivations and theory.

**Review Assessment: Checking Correctness Of Experiments:**

I assessed the sensibility of the experiments.

**Review Assessment: Thoroughness In Paper Reading:**

I read the paper at least twice and used my best judgement in assessing the paper.

---

> ### Author Response · Authors · 2019-11-11
> **Paper update and comments**
>
> We thank reviewer #3 for the extensive feedback. We have incorporated it as stated below.
>
> We have significantly improved the exposition in this paper, ameliorated the discussion of the related work and added the suggested new interesting references, added theorems and more formal statements in the main text, compressed the appendix and enhanced the description of the experimental setup.
>
> Proofs that the space "interpolates smoothly with curvature": we added formal proofs (see theorems 2 and 3) that all the operations are differentiable, i.e. the gradients are equal from both the left and right at 0, w.r.t. curvature, for the chosen models of hyperbolic and spherical geometry.
>
> k-addition definiteness in the spherical setting: we have added the formal condition that the k-addition be well-defined, and a proof that for two points this condition indeed recovers x != y / (k ||y||^2) - see Theorem 1.
>
> Theorem 5 shows that our proposed left-matrix-multiplication is an intrinsic averaging for Riemannian manifolds of constant curvature, i.e. it does commute with isometries when the matrix A is right-stochastic, which is not the case of right-matrix-multiplication. This is a desirable property for Riemannian vector averaging.
>
> Theorem 6 shows that weighted combinations in the tangent space used in the very recent works of [1,2] (appeared after ICLR submission deadline) is also an intrinsic averaging for Riemannian manifolds of constant sectional curvature, i.e. it does commute with isometries.
>
>
> Other comments:
> -Synthetic tree: contains |V| - 1 edges. We corrected this mistake.
> -Curvatures: are learned as we state in the paper section 4 and appendix F.
> -Citations: fixed.
> -Working with the Poincare ball as opposed to the hyperboloid model: this allows us to use gyrovector spaces which are defined either for the Poincare ball or the Klein model, as well as to connect those with the Riemannian geometry of the space. Moreover, as we show in the paper, we can now smoothly interpolate between all constant curvature spaces which is beneficial for learning curvatures without a priori deciding on their signs.
> -Statement about “general class of trees” replaced by “all weighted or unweighted trees”.
>
> [1] Hyperbolic Graph Neural Networks, I. Chami et al., Neurips’19
> [2] Hyperbolic Graph Convolutional Networks, Q. Liu et al., Neurips’19

---

> > ### Comment · AnonReviewer3 · 2019-11-14
> > **Response**
> >
> > Thank you for the response/revision. It clarifies all my questions

---

### Official Review · AnonReviewer2 · 2019-10-23
**Official Blind Review #2**

**Rating:** 6

**Review:**

In this paper, the authors address representation learning in non-Euclidean spaces.  The authors are motivated by constant curvature geometries, that can  provide a useful trade-off between Euclidean representations and Riemannian manifolds, i.e. arriving at more suitable representations than possible in the Euclidean space, while not sacrifising closed-form formulae for estimating distances, gradients and so on.

The authors point out that an extension of the gyrovector space formalization to spaces of constant positive curvature (spherical) is required, and with the corresponding formalization for hyperbolic spaces, one can arrive at a unified formalism that can interpolate smoothly between all geometries of constant curvature.  The authors propose to do so by replacing  the curvature while flipping the sign in the standard Poincare model.  This is a strong point reagarding this work, as it seems that no such unification has been attempted in the past (although simply replacing the curvature in the Poincare model seems a bit too straightforward to not have been attempted, it seems to be the case).

The authors also provide extensive supplementary material (around 20 pages) with detailed derivations and descriptions of experiments.  This also makes me wonder if this paper is more suitable for a journal - both in terms of the extensive supplementary material (e.g., curvature sampling algorithm and other details can be found only in supplementary), as well as the more rigorous review process that a journal paper goes through.

In the main paper, only proof of concept experiments are provided (one experiment), that nevertheless show competitive performance under varying settings.  However, it seems to me that such contributions in the rising field of geometric deep learning, where several challenges are yet to be overcome, can be beneficial for future research.

Since in the supplementary experiments also, it seems that curvature does have small variance in the results, how would the authors assess the robustness of the curvature sampling method with respect to the results?

**Experience Assessment:**

I have published one or two papers in this area.

**Review Assessment: Checking Correctness Of Derivations And Theory:**

I assessed the sensibility of the derivations and theory.

**Review Assessment: Checking Correctness Of Experiments:**

I assessed the sensibility of the experiments.

**Review Assessment: Thoroughness In Paper Reading:**

I read the paper thoroughly.

---

> ### Author Response · Authors · 2019-11-11
> **Thanks & paper update**
>
> We thank reviewer #2 for the useful feedback.
>
> We have significantly improved the exposition in this paper, ameliorated the discussion of the related work, added theorems and more formal statements in the main text,  compressed the appendix and enhanced the description of the experimental setup. We have updated the paper and kindly ask the reviewer to take another look.
>
> Robustness of the curvature sampling method: we provide confidence intervals in Table 2 of our results. We learn curvatures for each of the component spaces and show learned values together with confidence intervals in Appendix E. It can be seen that these variances are in the low regime.

---

### Public Comment · ~Jialin_Liu3 · 2019-11-06
**GCN's weight sharing mechanism understanding**

Really interesting work. GNN formed in Non-Euclidean Space is a promising topics.
I don't understand why the author said "The proposed architecture ... while this weight sharing can be understood as an efficient diffusion-like regularizer." in part 1 introduction, page 2. What does "diffusion-like regularizer" mean?

---

### Decision · Program_Chairs · 2019-12-19

**Decision:**

Reject

**Comment:**

This paper proposes using non-Euclidean spaces for GCNs, leveraging the gyrovector space formalism. The model allows products of constant curvature, both positive and negative, generalizing hyperbolic embeddings.

Reviewers got mixed impressions on this paper. Whereas some found its methodology compelling and its empirical evaluation satisfactory, it was generally perceived that this paper will greatly benefit from another round of reviewing. In particular, the authors should improve readability of the main text and provide a more thorough discussion on related recent (and concurrent) work.